# Human CD4-binding site antibody elicited by polyvalent DNA prime-protein boost vaccine neutralizes cross-clade tier-2-HIV strains

Shixia Wang [1], Kun-Wei Chan [2], Danlan Wei[3], Xiuwen Ma[1], Shuying Liu[4], Guangnan Hu[1], Saeyoung Park[1], Ruimin Pan[2], Ying Gu[5], Alexandra F. Nazzari[5], Adam S. Olia[5], Kai Xu[5], Bob C. Lin[5], Mark K. Louder [5], Krisha McKee[5], Nicole A. Doria-Rose [5], David Montefiori [6], Michael S. Seaman[7], Tongqing Zhou [5], Peter D. Kwong [5], James Arthos [3], Xiang-Peng Kong[2] & Shan Lu [1] ✉

The vaccine elicitation of HIV tier-2-neutralization antibodies has been a challenge. Here, we report the isolation and characterization of a CD4-binding site (CD4bs) specific monoclonal antibody, HmAb64, from a human volunteer immunized with a polyvalent DNA prime-protein boost HIV vaccine. HmAb64 is derived from heavy chain variable germline gene IGHV1-18 and light chain germline gene IGKV1-39. It has a third heavy chain complementarity-determining region (CDR H3) of 15 amino acids. On a cross-clade panel of 208 HIV-1 pseudo-virus strains, HmAb64 neutralized 20 (10%), including tier-2 strains from clades B, BC, C, and G. The cryo-EM structure of the antigen-binding fragment of HmAb64 in complex with a CNE40 SOSIP trimer revealed details of its recognition; HmAb64 uses both heavy and light CDR3s to recognize the CD4-binding loop, a critical component of the CD4bs. This study demonstrates that a gp120-based vaccine can elicit antibodies capable of tier 2-HIV neutralization.

While antiviral treatment has successfully halted disease progression in people with HIV-1[1], and various innovative preventive measures have slowed the global AIDS pandemic[2], an estimated 1.5 million new HIV infections occurred in 2021 worldwide[3]. Therefore, the development of a safe and effective HIV vaccine remains an important objective toward ending the HIV epidemic, as such a vaccine offers a low-cost solution to the global population, especially when other approaches are either expensive or have various side effects.

One of the bottlenecks to an effective HIV-1 vaccine is the difficulty in raising protective antibody responses, capable of neutralizing the diverse tier-2 neutralization-resistant viruses that typify HIV-1 transmission. Broadly neutralizing antibodies (bnAbs) capable of such neutralization have been a major focus of HIV-1 vaccine research for the past several decades, based on their broad cross-clade neutralization shown by in vitro assays[4–7] as well as their ability to protect monkeys in vivo from infection by simian-human chimeric virus challenge. VRC01, as one well-studied bnAb, was able to protect certain individuals from infection by HIV-1 viruses that were particularly sensitive to the antibody in AMP trial[8]. These antibodies have generally been identified only from HIV-1 infected donors, and a major challenge for the HIV vaccine field is the inability to elicit potent broadly

[1]Department of Medicine, University of Massachusetts Chan Medical School, Worcester, MA 01655, USA. [2]Department of Biochemistry and Molecular Pharmacology, New York University Grossman School of Medicine, New York, NY 10016, USA. [3]Laboratory of Immune Regulation, National Institute of Allergy and Infectious Diseases, NIH, Bethesda, MD 20892, USA. [4]SYL Consulting, Thousand Oak, CA 91320, USA. [5]Vaccine Research Center, National Institute of Allergy and Infectious Diseases, NIH, Bethesda, MD 20892, USA. [6]Department of Surgery, Duke University, Durham, NC 27710, USA. [7]Center for Virology and Vaccine Research, Beth Israel Deaconess Medical Center, Harvard Medical School, Boston, MA 02115, USA. ✉e-mail: shan.lu@umassmed.edu

neutralizing responses by vaccination, either in standard vaccine-test animals or human vaccine trials.

The HIV-1 envelope (Env) spike on the surface of virions comprises trimers of heterodimers formed by the two Env subunits, gp120 and gp41, which are the sole targets of virus-directed neutralizing antibodies. The HIV-1 infection process starts when gp120 binds to the CD4 receptor and co-receptors CCR5 or CXCR4 on host cells. Conformational changes induced by Env interaction with receptors trigger membrane fusion between the virus and host cell. Substantial immune evasion, including conformational masking[9], glycan shielding[10,11], and sequence variation[12], protect the HIV-1 Env spike from antibody-mediated neutralization. Nonetheless, remarkable progress has been made in isolating human monoclonal antibodies (mAbs) from HIV-infected patients with high breadth and potency, as well as the structural definition of the recognition of these antibodies on the HIV-1 Env spike[6,13–16].

Vulnerable sites on the Env trimer can be categorized into several major groups, each targeted by different categories of bnAbs, including (1) the CD4-binding site (CD4bs), (2) the V1V2 apex, (3) the V3-glycan, (4) the fusion peptide (FP), (5) the gp120–gp41 interface, and (6) the membrane-proximal external region (MPER)[17]. Among these sites, the CD4bs is particularly significant due to its role in mediating the initial step of virus–host interaction. This site is functionally conserved for interaction with the CD4 receptor, making it a critical target for vaccine design to induce antibodies to target CD4bs that can neutralize HIV-1[14,18].

Several classes of CD4bs-targeting bnAbs have been identified and characterized, including those derived from VH1−2 germline, such as the well-characterized VRC01 class of antibodies[4,6,16,19–21], those derived from the closely related VH1−46, such as 8ANC131[6,15], and

others like IOMA and similar antibodies[22]. Additional CD4bs antibodies have also been found to have derived from germline origins that are different from VH1−2 or VH1−46, and they utilize their heavy chain third complementarity determining regions (CDR H3s) to recognize the CD4bs; these types of antibodies have been isolated from multiple donors[21,23]. It is known that the breadth of cross-reactive neutralizing activity of bnAbs arises in some patients only after 2–4 years of continued HIV-1 infection[24,25], imposing a major challenge to HIV vaccines, which are generally delivered to humans in a few injections.

Here we report the isolation and characterization of a CD4bs-specific mAb, HmAb64, from a human volunteer who received a polyvalent gp120 DNA prime-protein boost HIV vaccine in a previously reported phase I study DP6-001[26,27]. This antibody is derived from a germline different from VH1−2 or VH1−46 and neutralizes several tier-2 resistant HIV-1 strains from different clades, albeit with moderate potency.

## Results

### Immunization of human volunteers and isolation of a novel CD4bs mAb

In our previously reported Phase I clinical trial (DP6-001) of a polyvalent gp120 DNA prime-protein boost HIV vaccine (PDPHV), healthy adult volunteers who were negative for HIV infection first received three immunizations of a DNA plasmid mixture, including five expressing gp120 antigens from diverse HIV-1 subtypes, followed by two protein immunizations of five recombinant gp120 proteins matching those in the DNA prime immunization (Fig. 1a). Peripheral blood collected at two weeks after the second protein immunization showed a high level of anti-gp120 IgG titers and cross-clade neutralizing antibody responses in volunteers' immune sera[26]. From DP6-

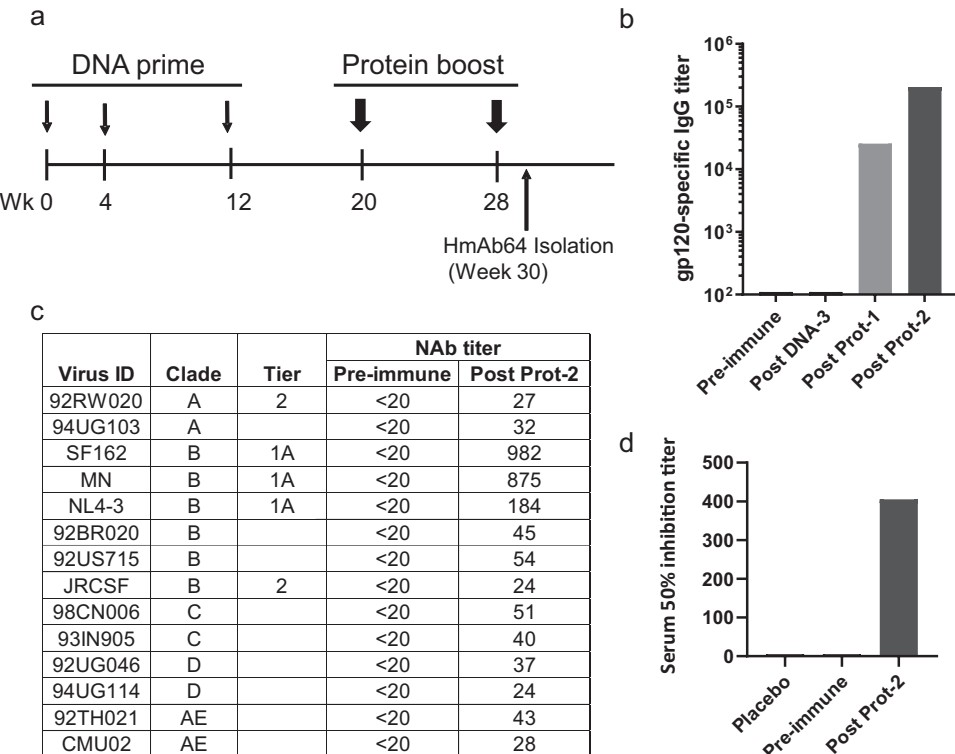

| Virus ID | Clade | Tier | NAb titer | |
| --- | --- | --- | --- | --- |
| | | | Pre-immune | Post Prot-2 |
| 92RW020 | A | 2 | <20 | 27 |
| 94UG103 | A | | <20 | 32 |
| SF162 | B | 1A | <20 | 982 |
| MN | B | 1A | <20 | 875 |
| NL4-3 | B | 1A | <20 | 184 |
| 92BR020 | B | | <20 | 45 |
| 92US715 | B | | <20 | 54 |
| JRCSF | B | 2 | <20 | 24 |
| 98CN006 | C | | <20 | 51 |
| 93IN905 | C | | <20 | 40 |
| 92UG046 | D | | <20 | 37 |
| 94UG114 | D | | <20 | 24 |
| 92TH021 | AE | | <20 | 43 |
| CMU02 | AE | | <20 | 28 |

**Fig. 1 | DNA prime and protein boost elicited serum HIV-1 Env specific neutralizing antibody responses targeting the CD4-binding site. a** Immunization scheme. Serum samples were collected 2 weeks after each immunization, and PBMCs were collected 2 weeks after the last immunization. **b** Development of gp120-specific IgG titers following key immunization steps. **c** Neutralization with the volunteer serum at peak level against a panel of pseudotyped viruses

expressing primary Env antigens from different subtypes of HIV-1 as indicated. **d** Virus capture competition assay using mAb b12 and volunteer sera. Serum 50% inhibition titer is reported as the reciprocal of serum dilution necessary to inhibit 50% of the virus captured by mAb b12. Source data are provided as a Source Data file.

001 study volunteers, a panel of gp120-specific human mAb were isolated, including those showing potent and cross-clade ADCC activities as previously reported[28]. In the current study, peripheral blood mononuclear cells (PBMC) collected from one DP6-001 volunteer (ABL009) were used to identify mAbs with potential neutralizing antibody activities. ABL009 was selected randomly for this pilot study. This volunteer's serum had increasing titers of gp120-specific IgG following the prime-boost immunization (Fig. 1b), and ABL009 serum showed low titer but broad neutralizing antibody activities against both the sensitive tier 1A virus and less sensitive viruses from subtypes A−E conducted by the Monogram assay (Fig. 1c). The neutralizing activities from ABL009 serum were similar to other DP6-001 volunteers as reported[26]. A cell-based competition assay further indicated the presence of CD4bs activities based on the ability of ABL009 serum to compete against a well-known CD4bs mAb b12 (Fig. 1d).

One mAb (HmAb64) was isolated from this volunteer which showed strong binding to the wild-type gp120 Env proteins from multiple HIV-1 isolates (JR-FL, AE, YU2) but not to their D368R mutant counterparts (Fig. 2a). Because CD4bs is lost in D368R mutants[29-31], our result suggested that HmAb64 targets at CD4bs of these gp120 proteins.

Defined by IMGT, VH and VL genes of HmAb64 were identified and assigned to germline IGHV1-18*01 and IGKV1-39*01. The CDRH3 length was 15 aa and the CDRL3 length was 9 aa according to Kabat definition. The overall mutation frequency from the germline for VH and VL was at 4.1% and 4.2%, respectively.

## The breadth of HmAb64's neutralizing activities

By using the standard TZM-bl assay, HmAb64 was shown to neutralize tier 1A and tier 1B HIV-1 primary isolates from subtypes B, C, AE, and AG (MN.3, SF162.LS, Bx08.16, 6535.3, MW965.26, CH0505.w4.3, 92BR025.9, DJ263.8, and TH023.6) (Table 1). At the same time, 14 HIV-1 isolates included in the same assay were not neutralized even at the maximum tested concentration (50 μg/ml). In a separate assay, HmAb64 was tested against the standard NIAID VRC panel of 208 HIV-1 viruses, and it was able to neutralize 20 viruses from diverse subtypes (10% of the panel) (Figs. 2b−d). Notably, 6 of the isolates were tier-2 or tier 1B/tier 2 neutralization resistant across clades B, BC, C, and G (Fig. 2c). Neutralization curves for tier 2 or tier1b/tier 2 neutralizers are included in Supplementary Fig 1.

## HmAb64 binding activity and CD4bs specificity

HmAb64 showed wide breadth against HIV-1 gp120 antigens from various subtypes. It demonstrated strong binding to the five autologous gp120 antigens (A2, B, Bal, Czm, and AE) used in the original PDPHV vaccine formulation (Fig. 3a, left panel). The binding affinity was observed in the low nanomolar range between $1.6 \times 10^{-8}$ M to $3.9 \times 10^{-8}$ M (Fig. 3b). HmAb64 was also able to bind to other unrelated (heterologous) gp120 antigens, though their levels of binding differ (Fig. 3a, right panel).

The binding of HmAb64 to HIV-1 virus was then verified in cells infected with HIV-1. HmAb64 was able to bind CD4+ T cells infected with HIV-1 isolate SF162 but not to uninfected cells, similar to the positive control VRC01, a well-characterized CD4bs mAb (Fig. 4a). The negative control human IgG did not exhibit such specific binding. This binding specificity was next tested using multiple human donors. HmAb64 was able to have consistent specific binding to SF162 infected CD4+ T cells from different human donors, and the binding level was similar to or higher than those of VRC01 (Fig. 4b).

The CD4bs specificity of HmAb64 to Env proteins from different HIV-1 isolates was further investigated in the presence of soluble CD4 (sCD4) in a Biacore-based assay. HmAb64 was able to bind to gp120 proteins from Bal or A244 viral isolates as well as BG505-SOSIP, but such binding was blocked by sCD4 in a dose-dependent manner, similar to VRC01. These results provided

further supporting data that HmAb64 is indeed a CD4bs-specific mAb (Fig. 5).

Furthermore, HmAb64 was able to block the binding of gp120 Env proteins from HIV-1 isolates to human CD4+ T cells in a FACS-based assay. The blocking by HmAb64 is similar in magnitude to that of Leu3A, a well-known CD4-directed mAb[32] (Supplementary Fig. 2a). The blocking was dose-dependent and could be observed with the dose of HmAb64 as low as 2 μg (Supplementary Fig. 2b). HmAb64 was able to have such a blocking effect on gp120 Env proteins from a wide diversity of HIV-1 isolates including SF162, Bal, AC02, AD8, CAP88, MW959, and CM244. The control Env from non-human primate virus SIV251 showed low-level binding, which could not be blocked by HmAb64 (Supplementary Fig. 2c), confirming the specificity of HmAb64 to the CD4bs of HIV Env proteins.

HmAb64 was also tested for its ability to bind to autoantigens such as dsDNA, ssDNA, LPS, and human insulin (Supplementary Fig. 3). It showed only minimal binding to these autoantigens, similar to the CD4bs-specific human mAb VRC01. In contrast, another human mAb 4E10, which is specific for gp41 of HIV-1 Env, had much higher binding to these autoantigens as previously reported[33], similar to several agents (ANA NC, ANA PC1, ANA, PC2) that are known to bind to autoantigens.

## Structure of HmAb64 in complex with HIV-1 Env

Structural characterization of HmAb64 was carried out by both protein crystallography and single-particle cryogenic electron microscopy (cryo-EM). The antigen-binding fragment (Fab) of HmAb64 was first crystallized, and its structure was resolved at 2.8 Å resolution (Supplementary Fig. 4a and Supplementary Table 1). The crystal structure revealed that HmAb64, in its apo form, has a relatively flat antigen binding site, like some known CD4bs bnAbs[21,31]. We then attempted to obtain a complex structure of HmAb64 Fab with a conformational stabilized Env trimer, CNE40 SOSIP.664[34-37]. However, this complex tended to form a honeycomb-like lattice structure (Supplementary Fig. 4b), likely induced by the interaction between HmAb64 Fabs, which introduced additional complications for the cryo-EM data analysis. This is likely HmAb64 specific as we have not observed such a phenomenon for other CD4bs mAbs. We, therefore, produced a single chain variable fragment (scFv) of HmAb64 and used it to form a complex with CNE40 SOSIP for cryoEM structure determination (Supplementary Fig. 5 and Supplementary Table 2).

The 3.7 Å cryo-EM structure revealed that each HmAb64 recognizes a gp120 protomer with its epitope region overlapping the CD4bs[21], resulting in a 3:1 scFv:Env trimer stoichiometry (Fig. 6a−c and Supplementary Fig. 6). Unlike other CD4bs bnAbs with their epitope regions compatible in the prefusion closed form Env trimer, HmAb64 recognition primarily occurred on the side of the CD4-binding loop closer to the trimer axis (Fig. 6c and Supplementary Fig. 7). This epitope engagement causes HmAb64 approach to the Env trimer in an open form conformation, similar to that of the open occluded form, with gp120 subunits rotating out from the center of the trimer, as typified by the trimers bound by CD4bs Abs b12 and Ab1303 (Fig. 6d and Supplementary Fig. 8)[38,39]. When superimposed on the gp41 HR1 helices, the HmAb64-bound gp41 aligned relatively well with that of Ab1303-bound (RMS displacement 2.6 Å and maximum displacement 5 Å), but the gp41 in the b12-bound Env showed a higher degree of conformational difference, especially in the HR2 region (RMS displacement 1.8 Å and maximum displacement 14 Å) (Supplementary Fig. 8). However, unlike the open occluded form with the V1V2 region positioned on top of gp120 subunits, the corresponding regions of V1V2 in our cryo-EM density map were not observed, indicating its disordered conformation, similar to those induced by CD4-binding, although we were unable to observe the densities of the four-stranded bridging sheet (Supplementary Figs. 6 and 7), a feature often present in other CD4-bound or asymmetric open form trimer structures[38,40-42].

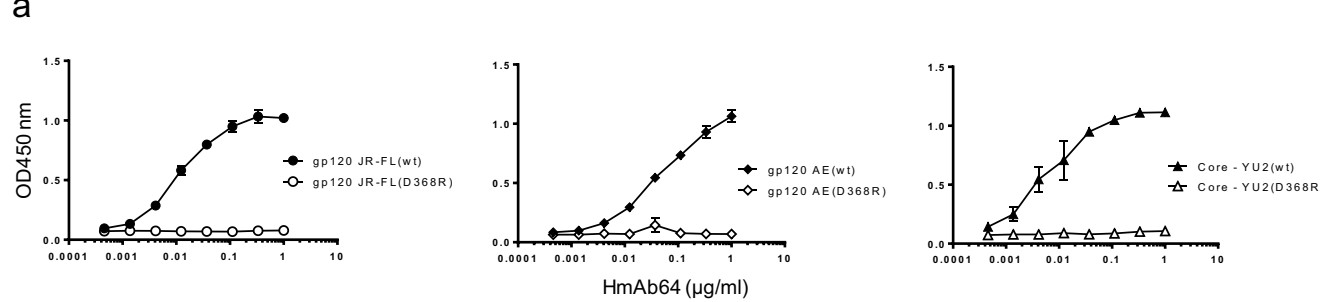

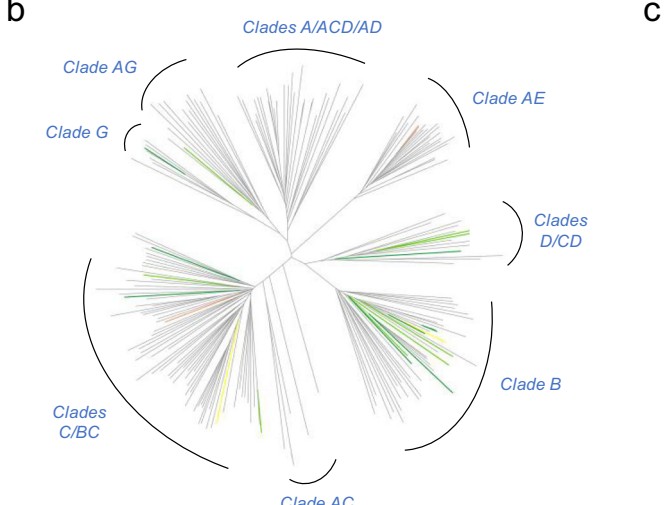

**c**

| Virus ID | Clade | HmAb64 | b12 | VRC01 | Tier |
|---|---|---|---|---|---|
| 6095.v1.c10 | ACD | 39.300 | 0.3250 | 0.7990 | ND |
| TH023.6 | AE | 0.077 | ND | 0.7710 | 1A |
| DJ263.8 | AG | 1.980 | >50 | 0.0540 | 1B |
| 6535.3 | B | 8.4900 | 0.8740 | 1.9700 | 1B/2 |
| ADA.DG | B | 17.4000 | 0.1280 | 0.4940 | ND |
| Bal.01 | B | 54.4000 | 0.0850 | 0.1400 | ND |
| BaL.26 | B | 28.2000 | 0.0640 | 0.0560 | 1B |
| BX08.16 | B | 14.2000 | 0.9240 | 0.4800 | 1B |
| CNE4 | B | 20.3000 | >50 | 0.4390 | ND |
| HXB2.DG | B | 0.3130 | 0.0010 | 0.0360 | 1B/2 |
| MN.3 | B | 7.2500 | <0.0006 | 0.0250 | 1A |
| SF162.LS | B | 2.0100 | 0.0320 | 0.2110 | 1A |
| CNE40 | BC | 0.1210 | 0.5530 | 0.2280 | 2 |
| 6644.v2.c33 | C | 2.3100 | 0.0310 | 0.1350 | 1B |
| BR025.9 | C | 15.3000 | >50 | 0.3590 | 1B/2 |
| MW965.26 | C | 0.0460 | 0.0020 | 0.0330 | 1A |
| ZM109.4 | C | 82.4000 | >50 | 0.1780 | 1B/2 |
| UG021.16 | D | 2.9600 | 0.8380 | 0.5740 | ND |
| UG024.2 | D | 4.1900 | >50 | 0.3500 | ND |
| X2131.C1.B5 | G | 76.3000 | ND | 0.6550 | 2 |

$IC_{50}$ (µg/ml)  | <0.001 | .001-.01 | .01-.100 | .100-1.00 | 1.00-10.0 | >10.0 |

**d**

| Neutralization resistance tiers | Viruses in VRC Panel | Neutralized by HmAb64 | % |
|---|---|---|---|
| Unclassified | 57 | 6 | 11% |
| Tier 1A | 5 | 4 | 80% |
| Tier 1B | 7 | 4 | 57% |
| Tier 1B/Tier 2 or Tier 2 | 127 | 6 | 5% |
| Tier 3 | 12 | 0 | 0 |
| Total | 208 | 20 | 10% |

**Fig. 2 | Vaccine-elicited antibody HmAb64 targets the CD4-binding site and neutralizes tier-2 HIV-1 strains. a** HmAb64 binding to gp120 Env proteins. HmAb64 showed strong binding to wild-type gp120 antigens from multiple HIV-1 isolates but not to their D368R mutants, indicating CD4-binding site specificity. Data are presented as mean values ± SEM ($n = 2$ duplicated wells). **b** Neutralization breadth. HmAb64 neutralized 20 viruses (10%) from diverse clades of the 208-strain NIAID panel. **c** Neutralization titers ($IC_{50}$) and tier designation of VRC panel viruses neutralized by HmAb64. **d** Number and percentage in each tier designation of VRC panel viruses neutralized by HmAb64. Source data are provided as a Source Data file.

This is likely due to the distinct binding mode that HmAb64 utilizes for gp120 engagement. Indeed, HmAb64 approached gp120 at an inward angle close to the Env 3-fold axis. In contrast, open occluded form conformation-recognized antibodies, such as b12 and Ab1303, as well as other CD4bs nAbs bind to gp120 at more outward angles from the interprotomer crevice (Supplementary Fig. 7). This distinct binding mode resulting in a different approach angle of HmAb64 from other CD4bs nAbs as well as CD4, offers new insight into subunit-based vaccine design. Our complex structure also revealed that most of the CDR loops of HmAb64 retain similar conformations as in the apo form except CDR H3, which has a shift of about 4.9 Å at the tip, and CDR L2 to fit its conformation into the pocket between the inner and outer domains of gp120 (Supplementary Fig. 4c). The overall epitope region of HmAb64 is extensive, with buried surface areas of 656 Å² and 561 Å², contributed by heavy and light chains, respectively (Fig. 6g). Among all the SHMs in the VH/VL regions of HmAb64, only two residues (Asn[L31] and Thr[L51]) were involved in direct contact with gp120 (Fig. 6h). This region almost completely covers CD4bs and partially overlaps with the epitope regions of other CD4bs nAbs (Fig. 6c, e), consistent with competition and binding analyses that HmAb64 is a CD4bs-specific Ab.

While HmAb64 employs a different approach to the CD4bs, showing a distinct binding mode from other CD4bs bnAbs and incompatibility with the closed Env trimer, its antigen recognition does exhibit a couple of features considered signatures for CD4bs bnAbs, as well as CD4 itself[21,31]. Firstly, a conserved interface region between the inner and outer domains of gp120, referred to as "the Phe43 cavity", named after the amino acid Phe43 of its primary receptor CD4, has been considered crucial for the binding of CD4bs bnAbs. HmAb64 engagement establishes extensive contact with this conserved region through its CDR H3 (Fig. 6e). Secondly, HmAb64 recognition involves an electrostatic interaction with the highly conserved negatively charged residue Asp[368] in the CD4 binding loop of gp120. Our cryo-EM complex structure revealed that residue Arg[H100e] in the CDR H3 loop of HmAb64, with its side chain pointing towards the CD4-binding loop, likely forms a salt bridge with Asp[368] (Fig. 6f). While the map densities

for the side chain of Asp[368] were not fully defined, the sequence substitution of Asp[368] resulted in a significant reduction in HmAb64 binding reactivity (Fig. 2a), suggesting its critical role in HmAb64 recognition. However, since HmAb64 was induced by the gp120 subunit vaccine, it is not unexpected that its binding to the trimer resulted in a conformational change in the local structure, such as the displacement of the β20/β21 loop of gp120 from its position as in both closed or open occluded form trimers. This displacement thereby pushes away the V1V2 region of gp120, explaining why the canonical four-stranded bridging sheet conformation observed in the CD4-bound open form trimer is not present in our HmAb64-bound trimer.

## Discussion

We have reported here that a CD4bs-specific mAb HmAb64 was isolated from a human volunteer who received an experimental HIV vaccine. The cryo-EM structure revealed HmAb64 recognition to include features that are considered signatures for CD4bs Abs as well as CD4 itself. Human CD4bs-specific mAbs were also isolated in the RV350 study from volunteers who received a late (6–8 years after the initial round of vaccination) boosting of RV144 vaccines with ALVAC-HIV and AIDSVAXgp120 B/E. These mAbs showed low-level neutralizing activities mainly against tier 1 viruses but also including one tier 2 virus[43]. No CD4bs mAbs have been reported from other human HIV vaccine studies in literature, including the RV144 study.

Neutralizing antibody function has been considered for several decades as the most important protective mechanism for a prophylactic HIV vaccine[17,44,45]. There are several well described regions on HIV Env serving as the main targets of neutralizing antibodies[17,45–48]. The CD4bs is considered one of the most critical targets as CD4bs specific mAbs isolated from HIV infected patients showed potent and broadly neutralizing activity[4,20,21,49,50]. At the same time, the CD4bs is a conformational epitope formed by non-linear amino acid sequences located at different regions of Env making it difficult to isolate and transplant to a scaffold immunogen. Significant effort has been devoted to the design and testing of immunogens to elicit CD4bs

**Table 1 | HmAb64 neutralizing activities (IC50 < 50 μg/ml) against HIV-1 pseudoviruses assayed at Duke University and BIDMC[a]**

| Virus | Subtype | Tier | IC50 (μg/mL) |
|---|---|---|---|
| MN.3 | B | 1A | 7.798 |
| SF162.LS | B | 1A | 7.107 |
| MW965.26 | C | 1A | 0.056 |
| TH023.6 | CRF01_AE | 1A | 0.166 |
| CH0505.w4.3 | C | 1A | 0.132 |
| Bx08.16 | B | 1B | 20.03 |
| 6535.3 | B | 1B | 33.32 |
| 92BR025.9 | C | 1B | 39.032 |
| DJ263.8 | CRF02_AG | 1B | 71.04 |

[a]The following viruses with IC50 > 50 ug/ml are not listed in the table: Q23.17 (clade A), BaL.26, 1056-10.TA11.1826, TRO.11 (clade B), 00836-2.5, ZM197M-PB7, 25710-2.43, Ce1176_A3, Ce703010217_B6, CH0505s (clade C), X1632_S2_B10 (clade G) and BJOX002000.03.2, CH119.10, CNE55 (CRF01 AE).

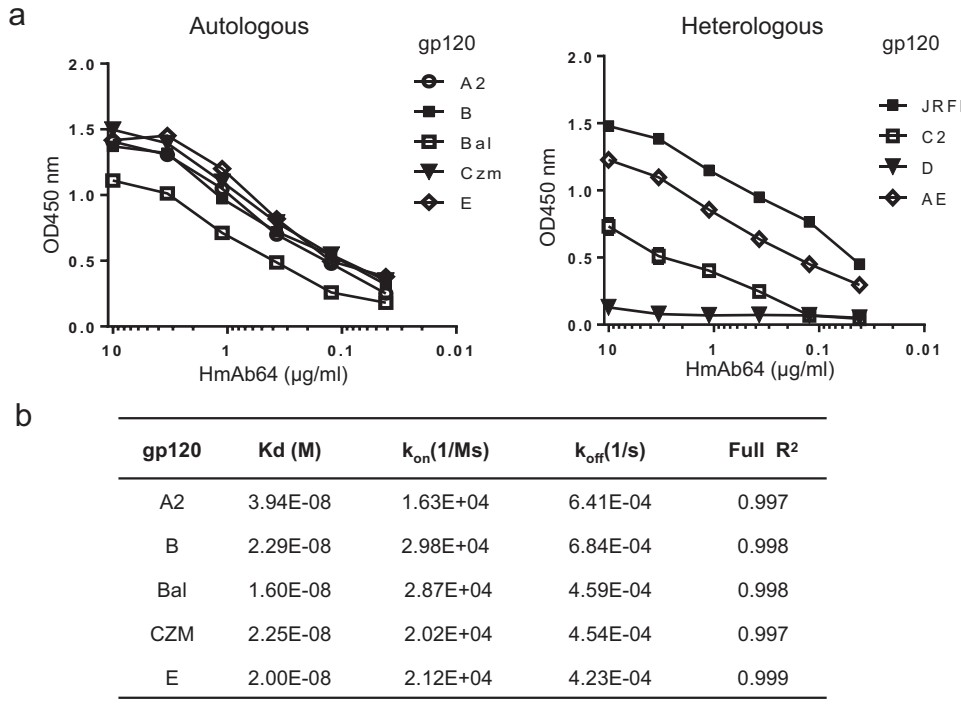

**Fig. 3 | Binding of HmAb64 to HIV-1 gp120 antigens. a** Binding to either autologous gp120 antigens included in the PDPHV vaccine (left) or heterologous gp120 antigens (right). **b** Biolayer interferometry binding kinetics of HmAb64 (Octet assay). Source data are provided as a Source Data file.

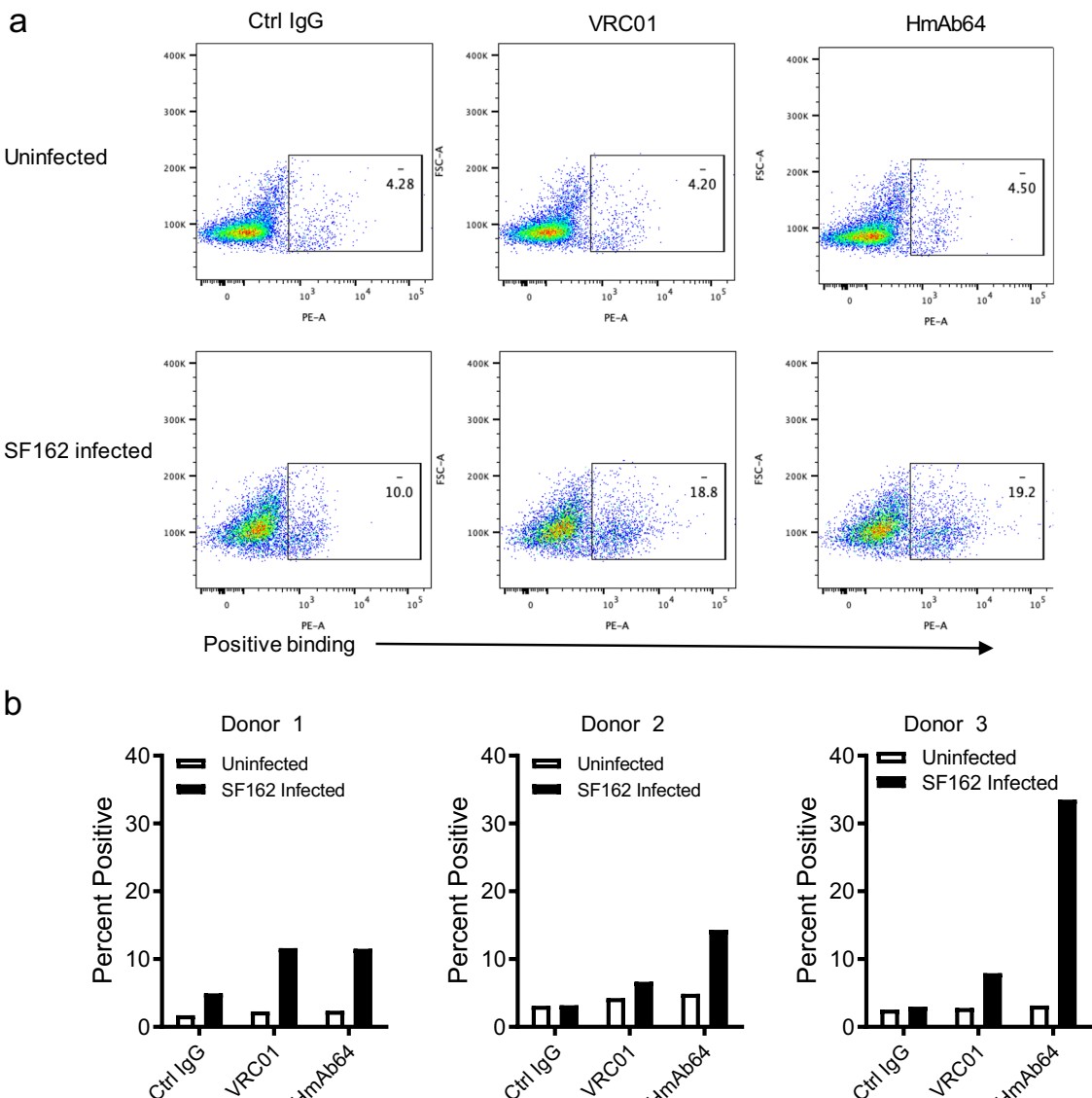

**Fig. 4 | Binding of HmAb64 to HIV-1 on cell surface. a** Examples of FACS analyses and gating showing the specific binding of HmAb64 to HIV-1 SF162 infected human CD4+ T cells but not to uninfected cells. CD4bs mAb VRC01 was used as the positive control, and normal human IgG was used as the negative control. **b** Percentage of HmAb64 binding to SF162 infected CD4+ T cells from three individual human donors. Source data are provided as a Source Data file.

antibodies, including germline targeting, which has not been successful in eliciting neutralizing antibodies, especially in human vaccine volunteer studies. Therefore, the success of isolating a cross-clade neutralizing CD4bs mAb from a healthy volunteer who received an HIV vaccine is a milestone in the development of HIV vaccines. HmAb64 was generated from a germline that is different from those for many known CD4bs bnAbs. Interestingly, among all the SHMs in the VH/VL regions of HmAb64, only two residues (Asn$^{L31}$ and Thr$^{L51}$) were involved in the gp120 contact (Fig. 6g), implying that there is less need for somatic mutation accumulation in inducing HmAb64-like Abs when applying germline-targeting strategy to guide an immune response.

This isolation of a CD4bs mAb from the PDPHV is not totally unexpected based on our previous studies. We first discovered that CD4bs antibodies were present in the sera of rabbits immunized by the DNA prime/protein boost but not those immunized with protein alone HIV vaccine[27] based on the serum competition assay against known CD4bs mAb b12. This finding supported our long-held belief that DNA immunization can deliver antigens with better conformation compared to the same antigen delivered in the form of a recombinant

protein. A nucleic acid vaccine like DNA vaccine may be particularly useful to preserve the conformation of CD4bs-related epitopes. This is because the in vivo production and folding of Env protein, along with proper post-translational modifications such as glycosylation, may better mimic the native Env structure than the recombinant Env protein produced and purified through in vitro processes.

Subsequently, we found that sera from volunteers in the DP6-001 clinical trial, who received the DNA prime-protein boost vaccine, can also compete against mAb b12[51], while those from volunteers in HVTN041 who were administered a recombinant gp120 protein alone vaccine exhibited a high level of V3 antibodies but no CD4bs antibody responses[51]. The successful isolation of CD4bs mAb HmAb64 from one of the DP6-001 volunteers provided more definitive evidence confirming the elicitation of CD4bs antibodies by the PDPHV vaccine. More significantly, the newly isolated HmAb64 was able to neutralize a panel of primary HIV isolates of diverse subtypes, with reasonable potency against some of them, although the overall percentage of breadth remains to be low compared to other typical CD4bs mAbs isolated from HIV-infected patients such as VRC01.

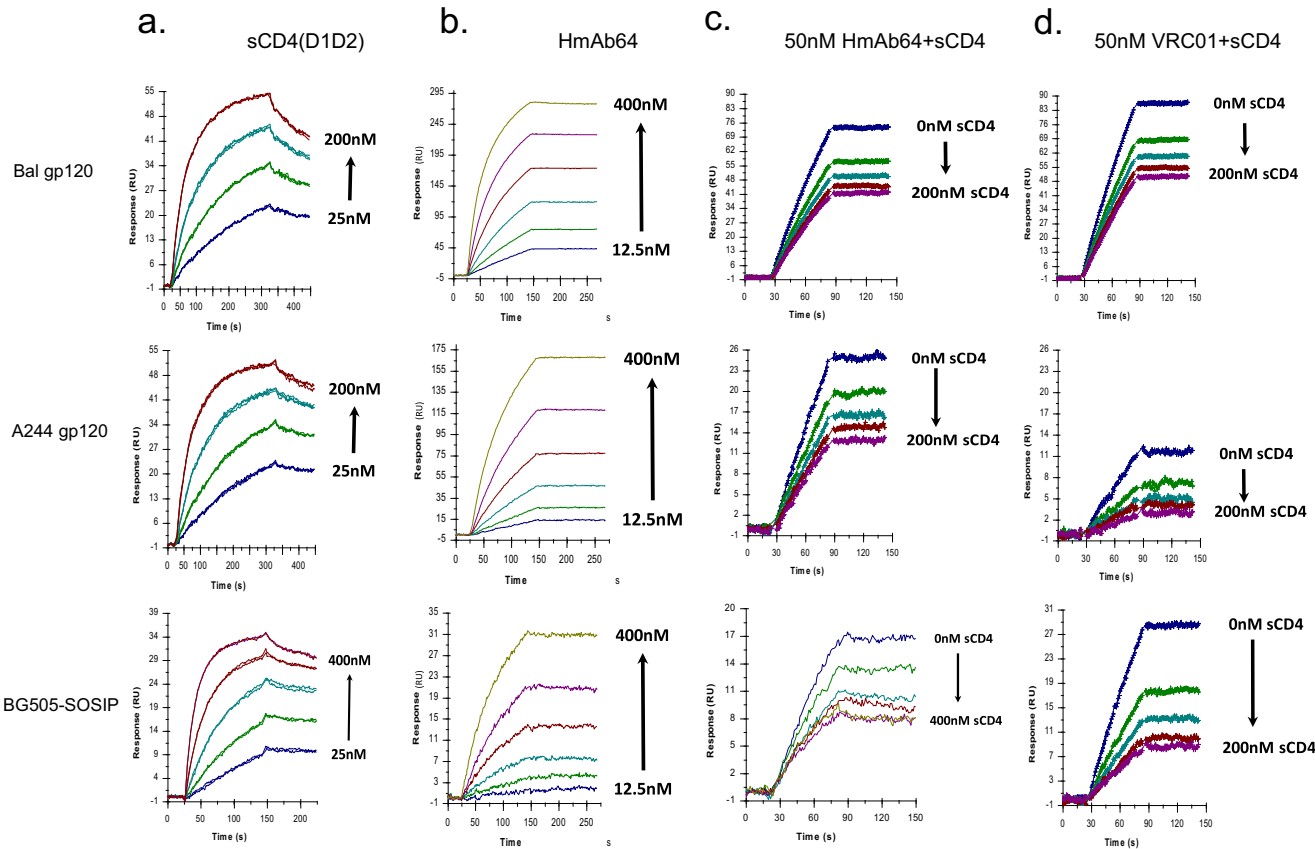

**Fig. 5 | Surface plasmon resonance assay of HmAb64 binding to HIV-1 Env antigens. a** Titration of sCD4 (D1D2 domain) binding to various HIV-1 Env antigens (Bal gp120, A244 gp120, and BG505-SOSIP). **b** Binding kinetics of HmAb64 to HIV-1 Env antigens. **c** Binding kinetics of HmAb64 to HIV-1 Env antigens in the presence of increasing amount of sCD4(D1D2). **d** Binding kinetics of VRC01 to HIV-1 Env antigens in the presence of increasing amount of sCD4(D1D2).

Our cryo-EM structure of the HmAb64 in complex with a SOSIP trimer has further revealed the structural basis of its function. The molecular details of the complex not only shows that the epitope of HmAb64 falls within the CD4 supersite previously identified[21], but also reveals the characteristics of antigen engagement of typical of CD4bs Abs, including a potential salt bridge interaction formed with $Asp^{368}$ and the hydrophobic interaction with $Phe^{43}$ cavity region of gp120. Moreover, the cryo-EM structure also provided a structural explanation of the neutralizing activity of HmAb64, i.e., its engagement of the HIV-1 Env trimer is more compatible with an open form conformation as its approach angle creates steric hindrance that prevents the formation of the bridging sheet, thus explaining why HmAb64 has limited breadth against more neutralizing resistant viruses.

In summary, we have demonstrated that neutralizing CD4bs antibodies with moderate neutralization potency can be induced in humans by vaccination in a relatively short time. Our structural analysis indicates that HmAb64 recognizes the CD4bs and regions compatible with an open-form conformation of the Env trimer. This recognition is distinct from previously identified CD4bs bnAbs, suggesting a strategy for including gp120 subunit as a part of HIV-1 vaccine targeting the induction of CD4bs Ab responses.

As learned from recent two antibody-mediated prevention (AMP) trials (HVTN704/HPTN085) even VRC01 alone did not prevent overall HIV-1 acquisition more effectively than placebo[8], suggesting a combo formulation of bNAbs may be needed. We need to further investigate whether more cross-subtype neutralizing antibodies are present against CD4bs and other targets in DP6-001 or recently completed HVTN124 study volunteers who received the second generation of PDPHV[52]. Ultimately, whether such polyvalent vaccine-induced cross-

subtype neutralizing antibodies can be protective needs to be tested in an efficacy study.

## Methods

### DP6-001 human sera and PBMCs
The serum and peripheral blood mononuclear cell (PBMC) samples from the DP6-001 trial (ClinicalTrials.gov Identifier: NCT00061243) were collected according to the institutional review board (IRB) approved protocol[26]. Volunteers signed informed consent to participate in the DP6-001 trial, including using human samples for further research studies. The serum and PBMC samples involved in the current study were from Subject 009, who received the HIV-1 DNA and protein vaccination comprised of full-length 120 Env antigens from HIV-1 clade A (92UG037.8), clade B (Bal and 92US715.6), clade C (96ZM652) and clade E (93TH976.17). The details of vaccine formulation, as well as clinical safety and immunogenicity results, were previously reported[26]. The serum samples used in this study were collected prior to immunization (Pre-immune), at 2 weeks after the 3rd DNA immunization (Post DNA-3, Week 14), 2 weeks after the 1st protein boost (Post Prot-1, Week 22), and 2 weeks after the 2nd protein boost (Post Prot-2, Week 30). The PBMCs used for Env-specific mAb isolation were collected at 2 weeks after the 2nd protein boost (Week 30).

### Memory B cell isolation and immortalization
The cryopreserved PBMCs from Subject 009 were thawed and cultured in RPMI 1640 with 15% Fetal Bovine Serum (FBS), 1% L-glutamine and 1% Penicillin-Streptomycin (P/S) at 37 °C with 5% $CO_2$ overnight. The B cells from overnight cultured PBMCs were first enriched using MACS Human B Cell Isolation Kit II (Miltenyi Biotec, San Diego, CA).

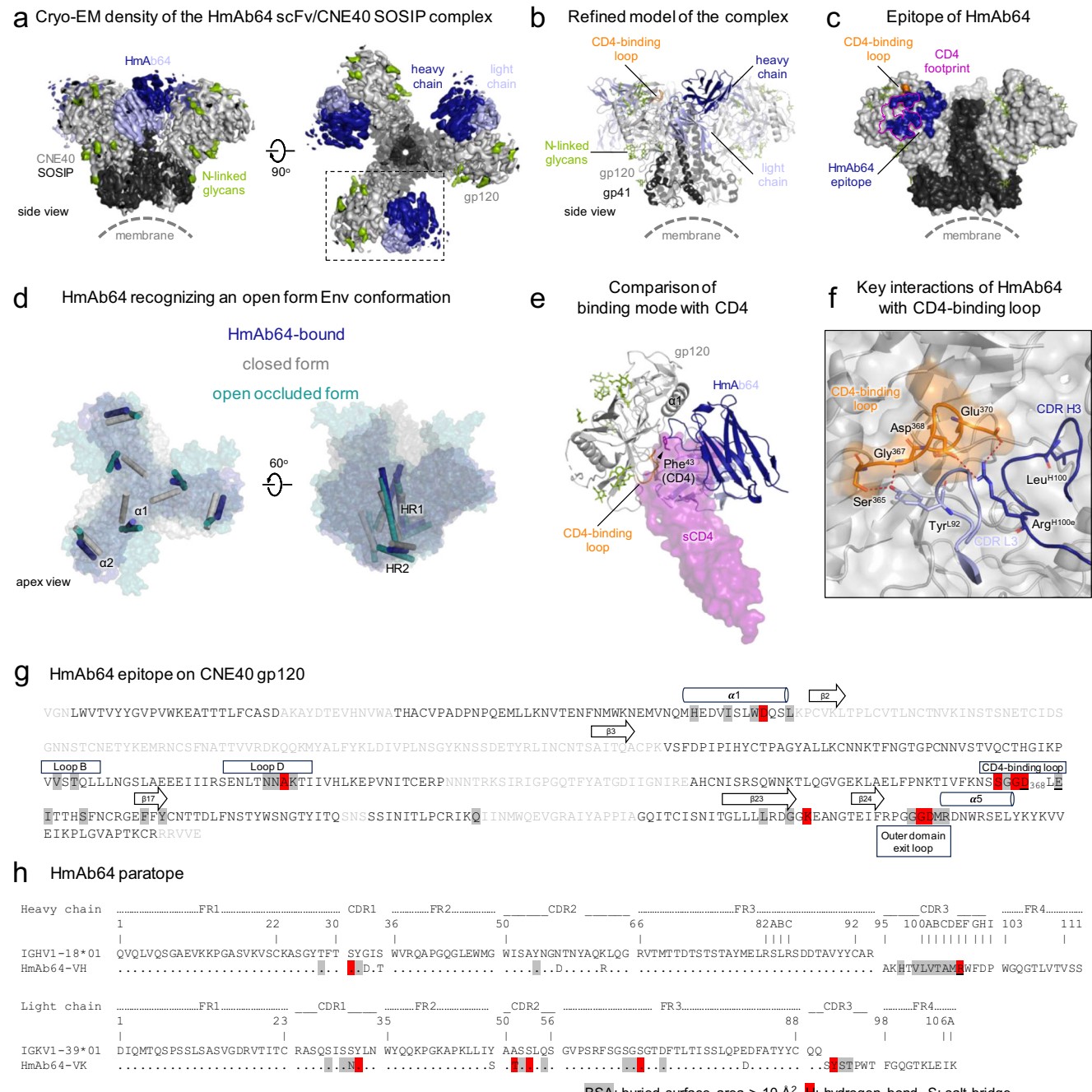

**Fig. 6 | Cryo-EM structure of HmAb64 in complex with a tier-2 HIV-1 CNE40 Env trimer. a** Cryo-EM density of HmAb64 in complex with CNE40 Env in two perpendicular views. The regions corresponding to gp120, gp41, and N-linked glycans, as well as heavy and light chains of bound HmAb64, were colored light gray, dark gray, green, blue, and light blue, respectively. **b** Refined cryo-EM structure of HmAb64 in complex with CNE40 Env with the same color usage as in (**a**) and the CD4-binding loop of gp120 highlighted in orange. **c** Surface representation of the HmAb64-bound Env trimer with HmAb64 epitope colored blue and the footprint of CD4 outlined in purple. **d** Comparison of the HmAb64-bound open form Env trimer (blue) with the closed form (gray) and the open occluded form trimer recognized by b12 (teal). Each gp120 and gp41 subunit was represented by two conserved helices, α1/α2, and HR1/HR2, respectively, in cylinders (also see Supplementary Fig. 8). **e** Superimposition of sCD4 (purple) onto an HmAb64-bound gp120 in the

same orientation as boxed in (**a**). The signature residue of CD4, Phe43, was highlighted in the sticks to indicate the approximate position of the Phe43 cavity. **f** Key residues in the CD4-binding loop and HmAb64 at the interface between gp120 and HmAb64. **g** Mapping of HmAb64 epitope on CNE40 gp120. Epitope residues were highlighted in gray shade, with those forming hydrogen bonds colored red and those forming salt bridges underlined. Amino acids that were disordered in the cryo-EM structure were shown in light gray font. Secondary structures of the antibody-binding regions were shown for reference. **h** Mapping of HmAb64 paratope. Paratope residues were highlighted in the gray shade, with those forming hydrogen bonds colored red and those forming salt bridges underlined. Only two SHM residues were involved in direct contact with gp120. The sequences were numbered according to the Kabat nomenclature.

Then, CD27+ memory B cells were isolated using the Human Memory B Cell Isolation Kit (Miltenyi Biotec). For EBV immortalization of B cells, the memory B cells isolated from Subject 009 were seeded at 5 cells per well in 96-well U-bottom microplates in 200 μl of complete RPMI medium containing 2.5 μg/ml CpG ODN2006, in the presence of EBV (30% supernatant of B95-8 cells) and irradiated allogeneic mono-nuclear cells (50,000 per well)[53]. After 12 days of culture at 37 °C with 5% CO2, 50 μl of culture supernatants were harvested and screened for gp120 binding by ELISA. The EBV-transformed cells from the gp120 binding positive wells were picked and expanded for clonal culture. The antibody from positive wells was further determined by ELISA with anti-lamda and anti-kappa secondary antibodies separately. The cells from gp120-specific clone #64 were harvested and preserved in RNA-later (Qiagen, Redwood City, CA) for RNA extraction and Ig gene cloning.

### Full-length immunoglobulin IgG heavy and Kappa chain gene isolation

RNA was extracted from clone #64 cells using RNeasy Mini Kit (Qiagen) and eluted in DNase/RNase-Free water (25 μl/million cells) for RT-PCR. The RT-PCR reactions were performed using the OneStep RT-PCR Kit (Qiagen). Briefly, for 25 μl reaction, the following reagents were added to each tube: 12.5 μl 2× Sensiscript RT rxn mix (Qiagen), 0.1 μl (4 units) of RNase inhibitor, 5 μl of #64 RNA sample, 0.5 μl of 5′ primer(s) and 0.5 μl of 3′ primer for heavy chain or kappa chain, and 6 μl of RNAase-free water.

For heavy chain RT-PCR, the 5′ primers were a mixture of 7 primers (VH1-Leader-A, B, C, D, E, F, and G, 5 nM/primer). The heavy chain 3′ primer was a single universal primer: CH2-IgG (20 nM). The RT-PCR reaction condition was at 50 °C for 30 min and 94 °C for 2 min, followed by 40 reaction cycles (94 °C for 15 s, 55 °C for 30 s and 68 °C for 100 s). For kappa chain RT-PCR: 5′ primers were a mixture of 2 primers (Vk1/2-Leader-A and B, 10 nM/primer). The Kappa chain 3′ primer was a single universal primer: CL2-kappa. The RT-PCR reaction condition was 50 °C for 30 min; 94 °C for 2 min, followed by 40 reaction cycles (94 °C for 15 s, 55 °C for 30 s, and 68 °C for 75 s). The PCR products for IgG heavy and kappa chains were analyzed by agarose gel electrophoresis. The 1.5 kb heavy chain and 0.7 kb kappa chain fragments were extracted using QIAquick PCR Microcentrifuge Protocol. Primers sequences can be found in Supplementary Table 3.

### Construction of heavy chain and Kappa chain expression plasmid

To construct the HmAb64 heavy and Kappa chains expression plasmids, PCR was conducted using HmAb heavy chain and light chain RT-PCR products as template. For heavy chain expression plasmid, VH1-LEADER-HindIII and CH2-IgG-NheI were used as 5′ primer and 3′ primer, respectively. For Kappa chain expression plasmid, Vk1/2-Leader-HindIII and CL2-kappa-BamHI were used as 5′ primer and 3′ primer, respectively. The 1.5 kb heavy chain fragment and the 0.7 kb kappa chain fragment were extracted from agarose gel and cloned into mammalian expression vector pJW4303 between HindIII and NheI cloning sites[54] and confirmed by DNA sequencing.

### Expression and production of mAb64 mAb

To express and produce the mAb64, heavy and light chain plasmids (mAb64-HC/pJW4303 and mAb64-KC/pJW4303, 1:1 ratio) were trans-fected into Freestyle 293 F cells (Invitrogen) as previously described[55]. At 2 days after transfection, the supernatant was harvested. The human IgG (HmAb64) antibody was purified with an AKTA FPLC system using Protein A HP columns (GE Healthcare).

### ELISA

ELISA was conducted to examine the HIV-1 Env-specific bindings in DP6-001 serum samples, EBV-transformed B cell media, and the final

mAb64 against gp120, gp120-Core, or their D368R mutant protein as previously described[26]. To detect the polyreactivity of antibodies, the ELISA plate was coated with ssDNA, dsDNA and LPS and Insulin with the final concentration of 10 μg/ml for ssDNA, dsDNA, and LPS, and 5 μg/ml for Insulin in PBS, 50 μl/well, overnight at RT. After blocking for 2 h at RT, properly diluted HmAb64 (50 μl/well) were incubated for 2 h at RT. HRP-conjugated goat anti-human IgG (Jackson, 109-035-098) in dilution buffer (1:1000, 50 μl/well) was incubated for 1 h at RT. At last, the plate was developed for 5 min at 37 °C in 100 μl of a 3,3′5,5′-tetramethylbenzidine substrate solution (Sigma). The reaction was stopped with 50 μl of 2 N $H_2SO_4$. Plates were read at 450 nm.

### Binding affinity analysis with Octet Qke

To evaluate the HmAb64 binding kinetics to different gp120 proteins, Octet Qke (ForteBio) assays were conducted based on biolayer inter-ferometry. Briefly, the HmAb64 antibody was loaded onto Protein G (ProG) sensor tips at 20 μg/ml in a kinetics buffer. After capture, bio-sensor tips were washed in a kinetics buffer, and a baseline measure-ment was recorded. The sensors were then incubated with serial diluted gp120 protein (600 nM–9.3 nM) to measure the association rate ($K_{on}$) and dissociation rate ($K_{off}$). The antibody binding kinetics and $K_d$ values ($K_{off}/K_{on}$) were analyzed by the ForteBio Data Analysis software package v7.1 using a 1:1 binding model.

### Neutralization assays

Two types of neutralization assays were performed to evaluate HmAb64 neutralizing activities.

(1) TZM-bl cell-based HIV-1 pseudovirus neutralization assays were performed using the previously validated protocol in TZM-bl cells[56] by the laboratories of Dr. David Montefiori at Duke University and Dr. Michael Seaman at Harvard Medical School. HIV-1 pseudoviruses were produced in HEK293T cells by co-transfection of a plasmid expressing gp160 envelope with the pSG3ΔEnv backbone vector. A panel of 23 pseudoviruses expressing HIV-1 Env was evaluated (Table 1). Serial dilutions of HmAb64 were performed in duplicate, followed by the addition of pseudovirus. Plates were incubated for 1 h at 37 °C followed by the addition of TZM.bl target cells ($1 \times 10^4$/well) and DEAE dextran (final concentration 11 μg/ml). Plates were incubated at 37 °C for 48 h and then developed with Promega Bright-Glo luciferase assay reagent per the manufacturer's instructions. Relative Light Units (RLUs) were collected by running plates on a Promega GloMax luminometer. Neutralization was calculated as a percent reduction in luciferase activity in the presence of antibodies compared to the luciferase activity induced by the virus control without antibodies, calculated as follows: [(1 − (average sample RLUs-cell control RLUs)/(virus control RLUs-cell control RLUs)] × 100.

(2) For the large standard panel of 208-Env pseudoviruses, neu-tralization assays were performed on a Beckman Coulter Biomek automated liquid handler integrated with Thermo Cytomat Incubator (37 °C and 5% CO₂) and Molecular Devices Spectramax Paradigm luminometer at the Vaccine Research Center, NIH as described previously[57]. The automated assay workflow miniaturized the pre-viously optimized HIV-1 neutralization assay system in the TZM-bl cells[56] in a 384-well plate format. The automated 384-well assay per-formed sample addition and dilution, virus addition with 45 min incubation, followed by TZM-bl cell addition and 48-h incubation, and the subsequent supernatant removal and addition of luciferase sub-strate followed by luminescence measurement.

### Virus capture competition

The ability of human immune sera to inhibit the capture of pseudo-virions by the CD4bs specific mAb b12 (NIH AIDS Reagent Program, Cat# 2640) was investigated using a previously described competition assay[27]. Pseudotyped virions in this assay expressing both HIV-1 Env and vesicular stomatitis virus G protein on the virion surface were

produced. The vesicular stomatitis virus G protein will mediate the entry of virions into the target cell line CF2.CD4.CCR5, irrespective of any neutralizing activity against the HIV envelope present in the sera, thus providing a sensitive readout of the captured virus as previously described[27]. The capture of pseudovirions by b12 will be blocked if there is a competing CD4bs antibody in the human sera. Specifically, microwells were coated with b12 (5 μg/ml) overnight and were then washed and blocked with 3% bovine serum albumin in PBS. Graded dilutions of human immune sera were added to the virus, and the virus-serum mixtures were then added to b12-coated ELISA wells for 3 h, followed by washing with PBS. CF2.CD4.CCR5 cells were overlaid, and 2 days later, infection was measured by assaying luciferase. The reciprocal serum dilution that inhibited b12-mediated virus capture by 50% was recorded.

## mAb binding to HIV-infected CD4+ T cells
Primary CD4+ T cells were isolated from PBMCs of healthy donors by negative selection (StemCell). Cells were cultured for 5 days in media containing anti-CD3, retinoic acid, and IL2 (20 μ/ml). On day 5, cultures were with 4 μl of HIV SF162 per $10^6$ cells (stock concentration 160 ng/ml). After 3 days, cells were stained with VRC01 (NIH AIDS Reagent Program, Cat# 12033, 2 μg) (positive controls), human IgG1 (negative control), and HmAb64 (2 μg), washed and incubated with 1 μl of goat anti-human-PE (Southern Biotech) and data collected on a FACSCanto (BD Biosciences) using standard protocols.

## HmAb64 blocking of gp120 binding to CD4+ T cells
The gp120s derived from HIV-1 AD8, AC02, CAP88, CM224, MW959, SF162, Bal, and SIVmac251 were biotinylated (Lynx biotinylation kit, Bio-Rad) and used to stain primary CD4+ T cells, isolated from healthy donors, followed by neutravidin PE (Invitrogen) in the absence or presence of the CD4 mAb Leu3a (positive control) or HmAb64, washed 2x and data was collected on a FACSCanto using standard protocols.

## Surface plasmon resonance analysis of CD4 or anti-Env mAb binding to surface immobilized BG505-SOSIP, Bal gp120, or A244 gp120
Experiments were performed using a Biacore 3000 instrument (Cytiva Life Sciences) using CM4 or CM5 sensor chips. The data were evaluated using BIA evaluation 4.1.1 software (Cytiva Life Sciences). The chip surface was activated by injecting 35 μl of a 1:1 mixture of 0.05 M N-hydroxysuccinimide and 0.2 M N-ethyl-N-(dimethylaminopropyl) carbodiimide at 5 μl/min. HIV gp120 or BG505-SOSIP at concentrations of 5 μg/ml in 10 mM NaOAc, pH 4.5, were immobilized to approximately 750 resonance units (RU). After the proteins were immobilized to the desired densities, unreacted sites on each surface were blocked with 35 μl of 1 M Tris-HCl (pH 8.0). One surface was activated and blocked without a ligand to act as a control surface for the non-specific binding of the soluble ligand. Any binding was subsequently subtracted from the remaining surfaces. The running buffer was HBS (pH 7.4), 3 mM EDTA, 0.005% Tween-20, 0.05% soluble carboxymethyl-dextran. Binding experiments were carried out at a flow rate of 25 μl/min at 25 °C. After a 2 min injection, the surface was washed for an additional 2 min in running buffer to follow the dissociation of the bound ligand from the surface. The surfaces were regenerated by multiple injections of 4.5 M $MgCl_2$ at a flow rate of 100 μl/min.

## Inhibition of anti-gp120 mAb binding to HIV-1 Envs by pre-bound sCD4 as measured by surface plasmon resonance analysis
Inhibition of anti-gp120 mAb binding to HIV-Env by sCD4 was performed by passing either buffer or sCD4 (D1D2) at the indicated concentrations over the surfaces described above at a flow rate of 25 μl/min at 25 °C, followed by a second sequential injection of a single concentration of the indicated mAb. The ability of sCD4 to inhibit the anti-gp120 mAb was determined by comparing the binding of the antibody in the absence or presence of the pre-bound sCD4 as described[58].

## Fab production and purification
The Fab fragment of HmAb64 was prepared by papain digestion as described previously[59] for crystal structure analysis. Briefly, the IgG molecule was mixed with papain (Worthington) at a 20:1 ratio in 100 mM Tris (pH 6.8) with 1 mM cysteine hydrochloride and 4 mM EDTA. The mixture was incubated for 1 h at 37 °C, and the reaction was stopped by adding 10 mM iodoacetamide. The Fab fragment was separated from Fc and the undigested IgG by a protein A column and further purified by size exclusion chromatography (SEC). The concentration of the Fab fragment for crystallization was 10 mg/mL.

## Crystallization, data collection, structure determination, and refinement
The preliminary crystals of HmAb64 Fab were obtained by robot screening using the vapor diffusion hanging drop method as previously described[60,61]. Crystals of HmAb64 Fab alone were obtained in a good solution of 14% polyethylene glycol 8000, 0.1 M HEPES pH 7.5, and 8% ethylene glycol. Crystals of HmAb64 Fab were first soaked in the mother liquor with an additional 20% glycerol (v/v) and 20% ethylene glycol, respectively, before being placed in the X-ray beam. Data of the crystals of HmAb64 Fab alone were collected at the synchrotron beamline X6A at the National Synchrotron Light Source (NSLS), Brookhaven National Laboratory. All data sets were processed using the HKL 2000 software package[62], and structures were determined by molecular replacement as we previously determined (PDB ID 4JO1)[60,63]. Cycles of refinement for each model were carried out in COOT and Phenix[64,65]. Figures were generated using PyMOL[66].

## Expression and production of scFv HmAb64
To aid cryo-EM studies of the HmAb64-Env complex while preventing aggregation caused by the antibody constant region, a single-chain Fv (scFv) version of HmAb64 was generated. This involved linking the VH and VL domains using 3× (Gly-Gly-Gly-Gly-Ser) linkers and adding a 6xHis-tag to the C-terminus. The resulting construct was cloned into the pVRC8400 vector. After expression in Expi293 cells, the protein was purified from the conditioned media using cOmplete His-Tag Purification Resin (Roche) via affinity purification. Further purification was then performed using SEC.

## Design and purification of CNE40 Env trimer and production of a scFv−HmAb64 complex
Single-chain Fc (scFc) tagged CNE40 Env trimer was generated by fusing the N-terminus of gp120 to knob-in-hole scFc domain, separated by an HRV3C site. The protein was expressed in FreeStyle™293 (ThermoFisher, Waltham, MA) cells at 37 °C for 5 days. The complex of scFv-HmAb64 and CNE40 Env trimer complex was produced by on-column HRV3C digestion. scFc-CNE40 supernatant was loaded on a protein A column and washed with PBS before adding purified scFv-HmAb64. Overnight incubation with HRV3C protease at 4 °C was performed to release the complex from the resin. The flowthrough was concentrated and further purified by SEC on a Superdex 200 Increase 10/300 GL column (GE Healthcare, Boston, MA) equilibrated in PBS.

## Cryo-EM sample preparation, data collection, and processing
A 3 μL of the scFv-HmAb64-CNE40 SOSIP protein complex, at 2.0 mg/mL, was applied onto a C-Flat 1.2/1.3 copper grid (Electron Microscopy Sciences, EMS), which had been freshly glow-discharged at 15 mA for 25 s using a PELCO easiGLOW system (TED PELLA). The sample grid was blotted with filter paper (Whatman) for 1.5 s at 4 °C under 100% relative humidity before plunging into liquid ethane using Vitrobot™ Mark IV (FEI).

Micrographs were collected at 105,000× nominal magnification with a pixel size of 0.426 Å on an FEI Titan Krios operated at 300 kV with a Gatan K3 imaging system. Zero-loss images were taken using an energy filter slit width of 20 eV. Movies were collected using Leginon[67] at a dose rate of 35.47 e⁻ Å⁻² s⁻¹ with a total exposure of 1.6 s, for an accumulated dose of 56.76 e⁻ Å⁻². Intermediate frames were recorded every 0.04 s for a total of 40 frames per micrograph. Due to a strong preferred orientation of the particles, a total of 16,962 micrographs were collected from 0°, 30°, 40°, and 45° tilted angles at a nominal defocus range of −0.5 to −2.5 μm.

Micrograph frames were aligned and dose-weighted using MotionCor2[68], and initial contrast-transfer-function (CTF) for each micrograph was estimated with CTFFIND4[69] in the Appion[70] platform for real-time data evaluation and pre-processing. Particle picking was trained in Warp[71]. Further processing, including 2D classification, 3D refinement, and map sharpening, was performed using cryoSPARC[72]. Briefly, a total of 1,859,663 warp-extracted particles were sorted through 2 rounds of reference-free 2D classification to remove junk containing false picks and denatured/dissociated particles (Supplementary Fig. 5). A total of 655,074 selected particles from the classes that exhibited Fv-bound trimer features were further sorted into four classes by heterogeneous refinement using an Ab-initio model low-pass filtered to 20 Å without importing symmetry (C1). The 259,148 particles belonging to the best class, which showed well-defined densities at the interface between gp120 and HmAb64, were subjected to 3D non-uniform (NU) refinement. Based on visual inspection, the reconstruction map exhibited a 3-fold symmetry. Therefore, the same set of 655,074 particles was re-processed with C3 symmetry imposed. Multi-class Ab-initio reconstruction (4 classes) was initially performed, followed by heterogeneous refinement (2 classes). The class of 316,221 particles was selected to conduct NU refinement with per-particle defocus and per-group CTF optimization. The resulting map reached an average resolution of 3.74 Å, based on the gold-standard Fourier shell curve (FSC) using a correlation cut-off of 0.143[73].

### Model building and refinement

To create an initial starting model for CNE40 SOSIP, the gp120 and gp41 subunits of a homologous SOSIP (PDB ID 6CK9[34]; chains G and B, respectively) were used as structural templates and replaced with the corresponding sequences in SWISS-MODEL[74]. Due to poor resolution at the regions corresponding to the trimer apex in the final 3D reconstruction map, the V1V2, V3, and C4 (β20/β21) domains of the initial gp120 model were truncated before being fitted into the map using ChimeraX[75]. This model, without scFv, was first refined using Phenix[76] by a single round of rigid body refinement, morphing, and simulating annealing, followed by rounds of manual inspection and model building in COOT[77]. The portion of HmAb64 scFv was then modeled using the crystal structure (Fv domains of PDB ID 6W73), by fitting VH and VL domains separately into the map in Chimera, followed by manual adjustment, particularly, of the loops at the interface regions. After multiple iterations of model building and real space refinement in COOT and Phenix, respectively, the glycan residues were added based on interpreted map densities using the Glyco module in COOT[78]. The final refinement was carried out by real space and B-factor refinement with secondary structure restraints using Phenix. Model validation was performed using MolProbity[79]. The Cα RMSD was calculated using the script available in the PyMOL wiki. All the figures were created using ChimeraX[75] or PyMOL[66].

### Reporting summary

Further information on research design is available in the Nature Portfolio Reporting Summary linked to this article.

## Data availability

The cryo-EM map and coordinates for the HmAb64 scFv-bound CNE40 SOSIP trimer have been deposited in the Electron Microscopy Data Bank (EMDB) and the Protein Data Bank (PDB) with the accession codes EMD-41569 and 8TR3, respectively. Source data are provided in this paper.

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

## Acknowledgements

We thank members of the Structural Biology Section and Structural Bioinformatics Core, Vaccine Research Center, for discussions or comments on the manuscript. We thank J. Baalwa, D. Ellenberger, F. Gao, B. Hahn, K. Hong, J. Kim, F. McCutchan, L. Morris, E. Sanders-Buell, G. Shaw, R. Swanstrom, M. Thomson, S. Tovanabutra, C. Williamson, and L. Zhang for contributing the HIV-1 envelope plasmids used in the 208-strain panel. We thank C. Moore, S. O'Dell, G. Padilla, S.D. Schmidt, C. Whittaker, and A.B. McDermott for assistance with neutralization assessments on the 208-strain panel. We thank Drs. W. Rice, B. Wang and A. Liang at the cryo-EM and microscopy core facilities in NYU Langone Medical Center for help with the cryo-EM data collection. X-ray diffraction data were collected at X6A NSLS, funded under GM-0080 and DOE from the United States, and we acknowledge members of the beamline for their help with data collection. This work was supported in part by the NIH NIAID HIV Vaccine Design and Development Teams contract N01 AI05394 (SL), and grants HIVRAD P01 AI082274 (SL), IPCAVD U19AI082676 (SL), R01 AI65250 (SL), R21/R33 AI087191 (SW), P01 AI100151 (XPK), and R01 AI145655 (XPK). Funding was provided in part by the Vaccine Research Center, an intramural Division of that National Institute of Allergy and Infectious Diseases, NIH.

## Author contributions

Conceptualization: S.W. and S.L. Investigation: S.W., K.W.C., D.W., X.M., G.H., S.Y.P., R.P., Y.G., A.F.N., A.S.O., K.X., B.C.L., M.K.L., K.M., N.A.D.R., D.M., M.S., and T.Z. Data analysis: S.W., K.W.C., S.Y.L., D.M., M.S., T.Z., J.A., X.P.K., and S.L. Resources: D.M., M.S., J.A., P.D.K., X.P.K, and S.L. Writing: S.W., S.Y.L., T.Z., P.D.K., X.P.K., and S.L. Supervision: P.D.K., J.A., X.P.K., and S.L.

## Competing interests

The authors declare no competing interests.
