## [Peer Review File · Nature Communications]

Human CD4-Binding Site Antibody Elicited by Polyvalent DNA Prime-Protein Boost Vaccine Neutralizes Cross-Clade Tier-2-HIV StrainsReviewers' Comments:

Reviewer #1:

Remarks to the Author:

The manuscript under review details the isolation and characterization of a monoclonal antibody, HmAb64, elicited by a polyvalent gp120-based vaccine regimen. Importantly, HmAb64 was isolated approximately 8 months after first immunization, and is reported to be capable of neutralizing several tier-2 HIV strains which challenges some preconceptions about the limitations of gp120 subunit-based vaccines.

Major Comments:

The authors summarize serum ID50 (Fig 1c) and mAb IC50 (Fig 2c) titers, but do not provide the neutralization curves or data. As the neutralization of Tier-2 envelopes is a central claim, I would consider it essential to provide the data that underpins this. This could be in the supplement, and reasonably could just include the Tier 2 viruses if space is a consideration.

Minor Comments:

The authors provide a somewhat shallow introduction. The manuscript would benefit from an introduction that better frames the importance of neutralizing CD4bs antibodies elicited by vaccination in humans.

The authors could better introduce broadly neutralizing antibodies as the current transition from Env-mediated entry to the epitopes targeted by bnAbs is jarring.

"were not neutralized at the IC50 concentration of 50 ug/ml". This should probably read, "were not neutralized even at the maximum tested concentration (50 ug/ml)

The authors refer to a Fig 2d that does not exist. This likely should be a reference to Fig. 2c. The reference to Fig. 2c should be Fig 2b.

The ELISA data should not be referred to to support differences in "binding affinities". Affinities should refer to the SPR (Fig 3b).

Fig. 2a and Fig. 5, both relate to the CD4bs specificity of the mAb. These data should be better consolidated. As the authors go on to provide a cryo-EM structure in complex, the mapping data could optionally be moved to the supplement.

The discussion begins with a convoluted priority claim that should be removed.

Reviewer #2:

Remarks to the Author:

Review of Wang et al. "Human CD4-Binding Site Antibody Elicited by Polyvalent DNA Prime-Protein Boost Vaccine Neutralizes Cross-Clade Tier-2-HIV Strains"

Wang et al. described a human mAb isolated by an HIV-negative volunteer after DNA prime boost vaccination that can neutralize a reasonably wide range of Tier 1B strains and a small handful of Tier 2 strains. While this is minor neutralization breadth in comparison to bnAbs from people living with HIV it is an important step in terms of what has been found in vaccinated humans so far. It is generally a carefully performed study that for the most part avoids the common pitfall of over exaggerating the breadth/potency of the mAb discovered. More information is required in the main text on why this particular individual was studied and the likelihood of similar antibodies in other people who received the same immunization.

However, the major weak point currently is the discussion, which does not fully explore the issues raised by this work, in particular is missing a thorough evaluation of how the dominant CD4bs targeting vaccine strategy, germ line targeting, can be informed by these results. Also, in contrast to the carefully pitched results section it overstates the breadth/potency of the mAb at times and places too much weight on the importance of DNA prime boost immunization in the generation of such neutralizing mAbs.

Specific criticisms

1. Abstract: "On a cross-clade panel of 208 HIV-1 pseudo-virus strains, HmAb64 neutralized 21 (10%), including tier-2 neutralization resistant strains from clades B, BC, C, and G."

Please clarify here how many tier 2 strains are neutralized as a number and a %

2. Page 3: "While antiviral treatment has successfully halted disease progression in many people infected with HIV-1"

Please use person first language throughout (people living with HIV-1)

3. Page 3: "Broadly neutralizing antibodies ... as their ability to protect monkeys in vivo from infection by simian-human chimeric virus challenge".

Please also discuss their impact in human studies to date.

4. Page 6: "ABL009 serum had increasing titers of gp120-specific IgG following the prime-boost immunization (Fig. 1b)".

Please include details of other participants in the figure. Is this response superior to theirs? is that why this individual studied of was it due to PBMC availability?

5. Page 6: "ABL009 serum showed low titer but broad neutralizing antibody activities against not only the sensitive Tier 1A virus but also less sensitive viruses from subtypes A, B, C, D and E conducted by the Monogram assay (Fig. 1c)."

Again, it's important how this compares to other patients as this is very low, please clarify in the text

6. Fig1D: Please write 50% inhibition titre on Y axis as it is confusing as is. Also the wording of legend needs improving as it's not at all clear how this assay works currently.

7. Page 7: "it was able to neutralize 21 viruses from diverse subtypes (10% of the panel) (Fig. 2c)."

Please state % tier 1B and tier 2. I found this misleading.

8. Page 8: "HmAb64 was also able to bind to other unrelated (heterologous) gp120 antigens, though their levels of binding affinities differ (Fig. 3a, right panel)."

Please state if the mAb neutralizes the corresponding PVs for all the gp120s it binds.

9. Page 8, 2nd Para: "The CD4bs specificity of HmAb64..."

I would have found this assay more informative if a tier 2 strain that it neutralizes had been used.

Please caveat that point (although of course the epitope stays the same regardless of strain)

10. Page 8, 3rd Para: "Furthermore, hmAb64..."

I don't find this that informative and would have preferred evidence of PBMC grown virus neutralization (say HXB2 as sensitive) to demonstrate that this mAb binds and neutralises Env as produced by CD4 T cells. I don't feel the data is strong enough to be its own main figure and please move to supplement.

11. Page 9: "However, this complex tends to form a mesh-like lattice structure (data not shown), likely induced by the interaction between the constant domain of HmAb64 Fab"

Does this likely play some role in function is this an HMAB64 specific thing or will any antibody binding to the CD4bs do this? Please comment in the text

12. Page 10: "similar to that recognized by b12 and other CD4-binding site antibodies". Please compare and contrast also to VRC01 as this point comes back in the discussion and is unclear currently.

13. Page 10: Please describe and reference Ab103 at first use

14. Page 10: "It is of note that the V1V2 apex was completely disordered (Fig. 7d, g)." Is this likely part of how the mAb neutralizes given the strain chosen is highly susceptible? Please comment in the text.

15. Page 12: "compatible with gp120 subunits rotating outward as in the open form conformation to minimize the steric hindrance from the neighboring gp120". Does this explain why it is good at tier1? Please clarify in the text.

16. Page 13: "Human CD4bs-specific mAbs were also isolated in RV350 study" Please specify if non-neutralizing or tier1 only neutralising

17. Page 13: "The CD4bs is considered the most critical as CD4bs specific mAbs isolated from HIV infected patients showed potent and broadly neutralizing activity" This is true of other bnAbs too... is the point here not that there is likely less mutational space for the virus as binding to CD4 must be maintained for infection? Please amend the text.

18. Page 13: "non-linear amino acid sequences located at different regions of Env making it a difficult immunogen to be produced by the conventional subunit protein vaccine designs." Please add discussion of how the conformational plasticity of Env is an extra challenge. i.e., that the more open form of Env or gp120 alone can trigger tier1 neutralisers but recognition of the CD4bs in the way that bnAbs do has been challenging

19. Page 13: "Significant effort has been devoted to the design and testing of novel immunogens to elicit CD4bs antibodies, which have not been successful in eliciting neutralizing antibodies, especially in human vaccine volunteer studies."

A much deeper analysis is needed here. Most of this work has been of late focused on germ line targeting approaches and how that ties into what has been observed here should be discussed.

20. Page 14: "That finding supported the long-held belief that DNA immunization can deliver antigens with better conformation comparing to the same antigen delivered in the form of a recombinant protein."

I think the role of DNA in this event are overstated. CD4bs non-neutralizing or tier 1 neutralizing antibodies have been induced by various immunogens including protein in animal models and that should be mentioned. The following paragraph goes on to refer back to mention the previous study where only V3 specific responses were mapped in the HVTN041 study which did not use DNA. However, it's unclear if the same protein component was used, if the total doses/timings were the same or how frequent the CD4bs response was across the different participants in the DNA prime study. It would be an ideal moment to refer back to the extra information I requested about why this particular individual was used to study mAbs in the results section and comment on the likely frequency and reproducibility of inducing hmAb64 type antibodies.

21. Page 15: "The new CD4bs mAb was able to neutralize a panel of primary HIV isolates of diverse subtypes and the neutralizing activities were quite potent against some of them. The total percentage of breadth against the standard panel is still quite moderate comparing comparing to other well-known CD4bs mAbs such as VRC01".

This is overstated and should be toned down. There are only 3 (?) tier 2s that are neutralized so while interesting it is extremely limited compared to VRC01, moderate doesn't really seem adequate.

22. Page 15: Re remainder of paragraph started with sentence in last comment, This argument does not really flow. It negates why identification of this antibody is important. Also, I think it is a bit of a stretch to jump straight from one mAb in one human to an efficacy/protection study. A better approach would be to estimate the level of this class of cross-neutralizer in existing participants at a serum level and repeat the experimental study to show reproducibility and then attempt to enhance by using immunogens targeted to this particular antibody subclass using the detailed structural information gathered in this paper.

23. Page 16: "we have demonstrated that authentic neutralizing CD4bs antibodies can be induced in human by vaccination in a relatively short time."

I don't understand the purpose of the word authentic here.

24. Page 16: "demonstrating that even when recognition of the CD4bs on gp120 does occur, clashes with regions surrounding the CD4bs preclude recognition of the prefusion-closed conformation of the Env trimer, the conformation recognized by most broadly neutralizing antibodies."

This is a crucial point: What does this suggest for vaccine design? Please expand on this.

Reviewer #3:

Remarks to the Author:

The manuscript entitled "Human CD4-Binding Site Antibody Elicited by Polyvalent DNA Prime-Protein Boost Vaccine Neutralizes Cross-Clade Tier-2-HIV Strains" by Wang et al. mainly focus to characterize a new monoclonal antibody, HmAb64. In this manuscript, authors claimed to isolate, and characteristics a new monoclonal antibody, HmAb64, which binds to the CD4-binding site, and the authors isolated this antibody from a human volunteer who was immunized by a polyvalent gp120 DNA prime-protein boost vaccine. In this manuscript, authors performed cloning, expression, purification, and crystal structure determination of HmAb64. Furthermore, they performed most of the biochemical and biophysical studies to characterize the binding of HmAb64 with gp120. Additionally, they performed the cell-killing assay to understand the potency of HmAb64. A significant part of this study was to elucidate the cryo-EM structure of gp120- HmAb64, and the authors performed cryo-EM to elucidate the 3D structure of the complex, which provides details about the HIV trimer and antibodies interactions.

The authors performed a huge amount of work in this manuscript, which covers cloning, protein biochemistry, cell biology, and biophysical work, like SPR, and finally used X-ray and cryo-EM to calculate the 3D structure. However, data needs to be represented appropriately. There are some flaws, which are described below.

After the antigen-antibody reaction, the author performed SEC and SDS PAGE, which they have mentioned in the method section; however, no data is presented even in the supplemental data file.

The authors mentioned that they tried to resolve a complex structure of HmAb64 Fab with an HIV Env trimer. However, they could not do so because this complex formed a mesh-like lattice structure (data not shown). It will be interesting to see a mesh-like lattice structure. Why is it forming a mesh-like lattice structure? Some other immunogen and antibody reactions also form network or helical lattice structures. It will be better to describe this portion also. So, readers can explore this also in the future.

The author has mentioned that "The HmAb64-bound gp120 subunits rotated outward comparing to those in b12- and ab1303-bound state (Fig. 7d)." However, from Figure 7D, it is not clear from figure how "gp120 subunits rotated outward comparing to those in b12- and ab1303-bound state". In that case, the author should incorporate a comparison of gp120 with b12 and gp120 with ab1303

separately. Additionally, the authors should compare the binding of HmAb64 with other existing antibodies. This comparison will help us to understand why HmAb64 is better than others and also help us to understand the mechanism neutralization of HmAb64.

From the figure, the prefusion close and occluded open forms are unclear.

The authors represented Figure 5, SPR data, which is very poorly described in the main text; Antigen-antibody interaction could be blocked by sCD4, and it worked in a dependent manner, which is evident from the figure. However, from the main text, it sounds like the authors presented their final conclusion, not describing their results.

Figure 7F shows the interaction of CDR L3 and CD4. Any interaction should be between the side chains of CDR L3 and CD4. However, the side chain densities of the fitted atomic models in the cryo-EM density maps must be shown adequately. It is difficult to confirm whether side chains are visible and whether there is any interaction. It is not very clear from the figure. From the 3D cryo-EM map, many areas are not appropriately resolved. Thus, no side chain densities were observed in the cryo-EM map. Are Asp368 and ArgH100e side chain density visible in the cryo-EM map? However, I am unable to see any extra density. Also, where is the cavity in Figure 7F?

No structure is shown after fitting the atomic model into the cryo-EM map. Hard to compare map Vs. Model fitting. Additionally, the authors mentioned in the method section that "Due to poor resolution at the regions corresponding to the trimer apex in the final 3D reconstruction map, the V1V2, V3, and C4 (β 20- β 21) domains of the initial gp120 model were truncated before being fitted into the map using UCSF Chimera v1.13.1 71." Thus, it is essential to show clearly which part is truncated and how was the fitting of the truncated part into the EM map. Definitely, the side chain should be visible at 3.7 Å, and it should be appropriately presented in the figures, at least in the interaction's region.

The outer part of the antibodies (dark blue) is not appropriately resolved; It appeared as some dust particles. Thus, it is possible that the outer region of the complex is flexible and difficult to resolve the high-res structure. Generally, that is the case for many of these types of viral protein complexes because they are highly glycosylated (spike-protein, Influenzas immunogen, HIV trimer), and the glycosylated part (the external part of spike, HIV trimer) is challenging to resolve at high resolution. In this current study, we can see that the outer part of the map is less resolved than the inner core part. Supplementary Figure 3, local resolution calculation shows authors achieved a high-resolution structure at the exterior region (cyan color 2Å), and the core is comparatively high resolution. Generally, it should be the opposite, and the core should be stable and able to reach high res. Generally, the core is more stable and easy to achieve high res, but authors achieved different results. Additionally, the map also shows that Fab density is weaker than the core. These two figures contradict each other and are difficult to understand. So, the authors should clarify this.

The authors should have mentioned how they calculate the local resolution in the method section. They should try local resolution estimation of unfiltered maps with ResMap.

At 3.7 Å resolution, some bulky side chains should be clear in the density map. Supplemental Figure 3d is the only figure where the authors represented a partial EM map fitted with the model. However, none of the side chains are clearly observed.

Why did authors collect data at "-" negative defocus values to "+" positive defocus values defocus range of -5.0 to 8.1 μm ? Also, the authors collected data using a dose rate of 35.47 $\text{e}^{-}\cdot\text{Å}^{-2}\cdot\text{s}^{-1}$ with a total exposure of 1.6 s, for an accumulated dose of 56.76 $\text{e}^{-}\cdot\text{Å}^{-2}$ over 40 frames. Thus, per frame exposure time is 0.04. Is this correct?

REVIEWER COMMENTS

Reviewer #1 (Remarks to the Author):

The manuscript under review details the isolation and characterization of a monoclonal antibody, HmAb64, elicited by a polyvalent gp120-based vaccine regimen. Importantly, HmAb64 was isolated approximately 8 months after first immunization, and is reported to be capable of neutralizing several tier-2 HIV strains which challenges some preconceptions about the limitations of gp120 subunit-based vaccines.

Major Comments:

The authors summarize serum ID50 (Fig 1c) and mAb IC50 (Fig 2c) titers, but do not provide the neutralization curves or data. As the neutralization of Tier-2 envelopes is a central claim, I would consider it essential to provide the data that underpins this. This could be in the supplement, and reasonably could just include the Tier 2 viruses if space is a consideration.

Response: As suggested by the reviewer, neutralization curves for HmAb64 against Tier-2 and Tier-1B/Tier-2 viruses are now provided as supplementary Fig. 1.

Minor Comments:

The authors provide a somewhat shallow introduction. The manuscript would benefit from an introduction that better frames the importance of neutralizing CD4bs antibodies elicited by vaccination in humans.

The authors could better introduce broadly neutralizing antibodies as the current transition from Env-mediated entry to the epitopes targeted by bnAbs is jarring.

Response: A revised Introduction is provided, where the transition to bnAb epitopes has been smoothed.

“were not neutralized at the IC50 concentration of 50 ug/ml”. This should probably read, “were not neutralized even at the maximum tested concentration (50 ug/ml)”

The authors refer to a Fig 2d that does not exist. This likely should be a reference to Fig. 2c. The reference to Fig. 2c should be Fig 2b.

Response:

- 1) “were not neutralized at the IC50 concentration of 50 ug/ml” is now revised to “were not neutralized even at the maximum tested concentration (50 ug/ml)”.
- 2) Sorry for the error and it now refers to Fig. 2c. And the reference to Fig 2c is changed to Fig. 2b-2d.

The ELISA data should not be referred to support differences in “binding affinities”. Affinities should refer to the SPR (Fig 3b).

Response: Thanks for pointing out this error. We corrected it in the result section

Fig. 2a and Fig. 5, both relate to the CD4bs specificity of the mAb. These data should be better

consolidated. As the authors go on to provide a cryo-EM structure in complex, the mapping data could optionally be moved to the supplement.

Response: Fig. 2a (mutation), Fig.5 (biacore CD4 competition) and Fig. 6 (T cell binding competition) provided important yet different supporting data to confirm the CD4bs specificity of HmAb64. In revision, we followed reviewer #2 suggestion to change the original Fig 6 to Supplementary Fig. 2.

The discussion begins with a convoluted priority claim that should be removed.

Response: Discussion is revised including the beginning sentence.

Reviewer #2 (Remarks to the Author):

Review of Wang et al. "Human CD4-Binding Site Antibody Elicited by Polyvalent DNA Prime-Protein Boost Vaccine Neutralizes Cross-Clade Tier-2-HIV Strains"

Wang et al. described a human mAb isolated by an HIV-negative volunteer after DNA prime boost vaccination that can neutralize a reasonably wide range of Tier 1B strains and a small handful of Tier 2 strains. While this is minor neutralization breadth in comparison to bnAbs from people living with HIV it is an important step in terms of what has been found in vaccinated humans so far. It is generally a carefully performed study that for the most part avoids the common pitfall of over exaggerating the breadth/potency of the mAb discovered. More information is required in the main text on why this particular individual was studied and the likelihood of similar antibodies in other people who received the same immunization.

Responses: We would like to thank reviewer #2's positive comments. All volunteers had comparable serum antibody responses as previously reported (reference 26 of revised manuscript). This volunteer (ABL-006) was selected purely randomly as a pilot study. We agree it is likely that similar antibodies may exist in other DP6-001 volunteers.

However, the major weak point currently is the discussion, which does not fully explore the issues raised by this work, in particular is missing a thorough evaluation of how the dominant CD4bs targeting vaccine strategy, germ line targeting, can be informed by these results. Also, in contrast to the carefully pitched results section it overstates the breadth/potency of the mAb at times and places too much weight on the importance of DNA prime boost immunization in the generation of such neutralizing mAbs.

Response: We have re-written the discussion based on this reviewer's comments. At the same time, our previous published work in rabbits did show that DNA prime immunization is critical in delivering conformation sensitive antigens and induced antibodies targeting CD4bs while the same antigens delivered by protein vaccine alone was not able do so (reference 27 of revised manuscript).

Specific criticisms

1. Abstract: "On a cross-clade panel of 208 HIV-1 pseudo-virus strains, HmAb64 neutralized 21 (10%), including tier-2 neutralization resistant strains from clades B, BC, C, and G."

Please clarify here how many tier 2 strains are neutralized as a number and a %

Response: We added a new Fig. 2d to provide the exact number and % of viruses neutralized by HmAb64 based on their tier designation in the VRC panel.

2. Page 3: "While antiviral treatment has successfully halted disease progression in many people infected with HIV-1"

Please use person first language throughout (people living with HIV-1)

Response: Changed to "people with HIV-1" throughout, as People with HIV (PWH) is the standard accepted way to refer to those that are HIV positive.

3. Page 3: "Broadly neutralizing antibodies ... as their ability to protect monkeys in vivo from infection by simian-human chimeric virus challenge".

Please also discuss their impact in human studies to date.

Response: Added the result of using VRC01 in human study AMP trial.

4. Page 6: “ABL009 serum had increasing titers of gp120-specific IgG following the prime-boost immunization (Fig. 1b)”.

Please include details of other participants in the figure. Is this response superior to theirs? is that why this individual studied of was it due to PBMC availability?

Response: As we previously already published, all volunteers in DP6-001 trial had comparable anti-gp120 IgG titers (reference 22). This volunteer was selected randomly as a pilot study. We add this information to the revised Result section,

5. Page 6: “ABL009 serum showed low titer but broad neutralizing antibody activities against not only the sensitive Tier 1A virus but also less sensitive viruses from subtypes A, B, C, D and E conducted by the Monogram assay (Fig. 1c).”

Again, it's important how this compares to other patients as this is very low, please clarify in the text

Response: We agree that the neutralizing antibody titers were low and it was noted in the original version. Other volunteers also had similar levels of Nab responses as previously reported (reference 22). We have now indicated this in revised manuscript as suggested.

6. Fig1D: Please write 50% inhibition titre on Y axis as it is confusing as is. Also the wording of legend needs improving as it's not at all clear how this assay works currently.

Response: We agree and have revised the labeling on Y-axis and the legend for Fig. 1d.

7. Page 7: “it was able to neutralize 21 viruses from diverse subtypes (10% of the panel) (Fig. 2c).” Please state % tier 1B and tier 2. I found this misleading.

Response: We now added % information as Fig 2d.

8. Page 8: “HmAb64 was also able to bind to other unrelated (heterologous) gp120 antigens, though their levels of binding affinities differ (Fig. 3a, right panel).”

Please state if the mAb neutralizes the corresponding PVs for all the gp120s it binds.

Response: For neutralizing antibody assays conducted at VRC or Duke University, no PVs available expressing the gp120 antigens as described in Fig 3a.

9. Page 8, 2nd Para: “The CD4bs specificity of HmAb64...”

I would have found this assay more informative if a tier 2 strain that it neutralizes had been used. Please caveat that point (although of course the epitope stays the same regardless of strain)

Response: We agreed with reviewer’s point. However, this experiment was conducted first to confirm CD4 binding specificity before VRC panel neutralization assay.

10. Page 8, 3rd Para: “Furthermore, hmAb64...”

I don’t find this that informative and would have preferred evidence of PBMC grown virus neutralization (say HXB2 as sensitive) to demonstrate that this mAb binds and neutralises Env as produced by CD4 T

cells. I don't feel the data is strong enough to be its own main figure and please move to supplement.

Response: We now changed the original Fig. 6 to Supplementary Fig. 2.

11. Page 9: "However, this complex tends to form a mesh-like lattice structure (data not shown), likely induced by the interaction between the constant domain of HmAb64 Fab"

Does this likely play some role in function is this an HMAB64 specific thing or will any antibody binding to the CD4bs do this? Please comment in the text

Response: The mesh-like lattice structure is likely caused by the interaction between the constant domain of the Fab molecule of HmAb64, as using scFv eliminated the issue. We have now included a panel in Supplementary Figure 4 (Supplementary Fig. 4b) to illustrate this point and have cited it in the text. We have also added a sentence in the text to specify that this phenomenon is specific to HmAb64.

12. Page 10: "similar to that recognized by b12 and other CD4-binding site antibodies". Please compare and contrast also to VRC01 as this point comes back in the discussion and is unclear currently.

Response: We have added a new figure, Supplementary Figure 7, to compare HmAb64's interaction with gp120 with that of several other CD4bs mAbs in details as well as CD4 itself. This figure is now greatly clarified the differences of the approach angle of HmAb64 to the CD4bs from that of other CD4bs Abs, such as b12 and VRC01.

13. Page 10: Please describe and reference Ab103 at first use

Response: We have now mentioned Ab103 in the text and cited the references when it's first mentioned.

14. Page 10: "It is of note that the V1V2 apex was completely disordered (Fig. 7d, g)."

Is this likely part of how the mAb neutralizes given the strain chosen is highly susceptible? Please comment in the text.

Response: We have now revised the text carefully to discuss the unique binding mode of HmAb64 where the bridging sheet and V1V2 were displaced, and also provide some text in the Discussion to suggest that this binding mode may be the origin of its limited neutralization breadth.

15. Page 12: "compatible with gp120 subunits rotating outward as in the open form conformation to minimize the steric hindrance from the neighboring gp120".

Does this explain why it is good at tier1? Please clarify in the text.

Response: We have now added text to include this suggestion.

16. Page 13: "Human CD4bs-specific mAbs were also isolated in RV350 study"

Please specify if non-neutralizing or tier1 only neutralising.

Response: These mAbs showed low level neutralizing activities mainly against Tier 1 viruses but also including one Tier 2 virus and we added this information to revised Discussion section.

17. Page 13: "The CD4bs is considered the most critical as CD4bs specific mAbs isolated from HIV

infected patients showed potent and broadly neutralizing activity”

This is true of other bnAbs too... is the point here not that there is likely less mutational space for the virus as binding to CD4 must be maintained for infection? Please amend the text.

Response: The reviewer is correct and we amended the text.

18. Page 13: “non-linear amino acid sequences located at different regions of Env making it a difficult immunogen to be produced by the conventional subunit protein vaccine designs.”

Please add discussion of how the conformational plasticity of Env is an extra challenge. i.e., that the more open form of Env or gp120 alone can trigger tier1 neutralisers but recognition of the CD4bs in the way that bnAbs do has been challenging.

Responses: We have revised the discussion based on reviewer’s comments.

19. Page 13: “Significant effort has been devoted to the design and testing of novel immunogens to elicit CD4bs antibodies, which have not been successful in eliciting neutralizing antibodies, especially in human vaccine volunteer studies.”

A much deeper analysis is needed here. Most of this work has been of late focused on germ line targeting approaches and how that ties into what has been observed here should be discussed.

Responses: HmAb64 is from a germline different from those of known bnAbs. We added this observation in the discussion.

20. Page 14: “That finding supported the long-held belief that DNA immunization can deliver antigens with better conformation comparing to the same antigen delivered in the form of a recombinant protein.”

I think the role of DNA in this event are overstated. CD4bs non-neutralizing or tier 1 neutralizing antibodies have been induced by various immunogens including protein in animal models and that should be mentioned. The following paragraph goes on to refer back to mention the previous study where only V3 specific responses were mapped in the HVTN041 study which did not use DNA. However, it’s unclear if the same protein component was used, if the total doses/timings were the same or how frequent the CD4bs response was across the different participants in the DNA prime study. It would be an ideal moment to refer back to the extra information I requested about why this particular individual was used to study mAbs in the results section and comment on the likely frequency and reproducibility of inducing hmAb64 type antibodies.

Responses: As we showed in a rabbit study report, when the same gp120 antigens were delivered side by side between protein alone and DNA prime-protein boost, there were clear CD4bs binding antibodies elicited from the DNA prime-protein boost group but not the protein alone group (reference 27 of revised manuscript).

21. Page 15: “The new CD4bs mAb was able to neutralize a panel of primary HIV isolates of diverse subtypes and the neutralizing activities were quite potent against some of them. The total percentage of breadth against the standard panel is still quite moderate comparing to other well-known CD4bs mAbs such as VRC01”.

This is overstated and should be toned down. There are only 3 (?) tier 2s that are neutralized so while interesting it is extremely limited compared to VRC01, **moderate** doesn’t really seem adequate.

Response: A total of 7 Tier 2 or Tier 1b/Tier 2 viruses were neutralized. We changed the word to “low”.

22. Page 15: Re remainder of paragraph started with sentence in last comment, This argument does not really flow. It negates why identification of this antibody is important. Also, I think it is a bit of a stretch to jump straight from one mAb in one human to an efficacy/protection study. A better approach would be to estimate the level of this class of cross-neutralizer in existing participants at a serum level and repeat the experimental study to show reproducibility and then attempt to enhance by using immunogens targeted to this particular antibody subclass using the detailed structural information gathered in this paper.

Response: We are planning to isolate similar mAbs from more volunteers as suggested by the reviewer.

23. Page 16: “we have demonstrated that authentic neutralizing CD4bs antibodies can be induced in human by vaccination in a relatively short time.”
I don't understand the purpose of the word authentic here.

Response: We have removed the word “authentic”.

24. Page 16: “demonstrating that even when recognition of the CD4bs on gp120 does occur, clashes with regions surrounding the CD4bs preclude recognition of the prefusion-closed conformation of the Env trimer, the conformation recognized by most broadly neutralizing antibodies.”
This is a crucial point: What does this suggest for vaccine design? Please expand on this.

Response: We have now added some speculation how the data of HmAbH64 might impact on vaccine design.

Reviewer #3 (Remarks to the Author):

The manuscript entitled “Human CD4-Binding Site Antibody Elicited by Polyvalent DNA Prime-Protein Boost Vaccine Neutralizes Cross-Clade Tier-2-HIV Strains” by Wang et al. mainly focus to characterize a new monoclonal antibody, HmAb64. In this manuscript, authors claimed to isolate, and characteristics a new monoclonal antibody, HmAb64, which binds to the CD4-binding site, and the authors isolated this antibody from a human volunteer who was immunized by a polyvalent gp120 DNA prime-protein boost vaccine. In this manuscript, authors performed cloning, expression, purification, and crystal structure determination of HmAb64. Furthermore, they performed most of the biochemical and biophysical studies to characterize the binding of HmAb64 with gp120. Additionally, they performed the cell-killing assay to understand the potency of HmAb64. A significant part of this study was to elucidate the cryo-EM structure of gp120- HmAb64, and the authors performed cryo-EM to elucidate the 3D structure of the complex, which provides details about the HIV trimer and antibodies interactions.

The authors performed a huge amount of work in this manuscript, which covers cloning, protein biochemistry, cell biology, and biophysical work, like SPR, and finally used X-ray and cryo-EM to calculate the 3D structure. However, data needs to be represented appropriately. There are some flaws, which are described below.

After the antigen-antibody reaction, the author performed SEC and SDS PAGE, which they have mentioned in the method section; however, no data is presented even in the supplemental data file.

Response: We have revised the method section to remove any assay that no data are included.

The authors mentioned that they tried to resolve a complex structure of HmAb64 Fab with an HIV Env trimer. However, they could not do so because this complex formed a mesh-like lattice structure (data not shown). It will be interesting to see a mesh-like lattice structure. Why is it forming a mesh-like lattice structure? Some other immunogen and antibody reactions also form network or helical lattice structures. It will be better to describe this portion also. So, readers can explore this also in the future.

Response: As mentioned in one of our Responses to Reviewer 2’s comments, we have now included a panel in Supplementary Figure 4 (Supplementary Fig. 4b) showing the 2D mesh-like lattice structure observed under EM. The details from the 2D clarification images clearly showed that the mesh-like structure was caused by the interactions between the Fab molecules.

The author has mentioned that “The HmAb64-bound gp120 subunits rotated outward comparing to those in b12- and ab1303-bound state (Fig. 7d).” However, from Figure 7D, it is not clear from figure how “gp120 subunits rotated outward comparing to those in b12- and ab1303-bound state”. In that case, the author should incorporate a comparison of gp120 with b12 and gp120 with ab1303 separately. Additionally, the authors should compare the binding of HmAb64 with other existing antibodies. This comparison will help us to understand why HmAb64 is better than others and also help us to understand the mechanism neutralization of HmAb64.

Response: We have modified the corresponding panel, now Fig. 6d, and have added additional figures, Supplementary Figures 7 and 8, to compare the overall rotation of gp120 and gp41 subunits in the closed form, HmAb64-bound, and b12-bound states (now Fig. 6d), and with the Ab1303-bound state (Supplementary Fig. 7 and 8). Due to the missing of the apical regions (V1V2, V3 as well as partial C4) of gp120 in the HmAb64-bound model, we only showed the conserved helices $\alpha 1$ and $\alpha 2$ to represent the

gp120 subunits, and HR1 and HR2 to represent the gp41 subunits. In addition, because of the varied ranges of trimer opening, instead of directly superimposing the whole trimer with their bound antibodies, we have provided a structural comparison of one gp120 protomer bound with HmAb64 and other CD4bs nAbs side by side (Supplementary Figure 7).

From the figure, the prefusion close and occluded open forms are unclear.

Response: The figure (now Fig. 6d) has been modified to make it clear.

The authors represented Figure 5, SPR data, which is very poorly described in the main text; Antigen-antibody interaction could be blocked by sCD4, and it worked in a dependent manner, which is evident from the figure. However, from the main text, it sounds like the authors presented their final conclusion, not describing their results.

Response: The wording in this section was revised as suggested by the reviewer.

Figure 7F shows the interaction of CDR L3 and CD4. Any interaction should be between the side chains of CDR L3 and CD4. However, the side chain densities of the fitted atomic models in the cryo-EM density maps must be shown adequately. It is difficult to confirm whether side chains are visible and whether there is any interaction. It is not very clear from the figure. From the 3D cryo-EM map, many areas are not appropriately resolved. Thus, no side chain densities were observed in the cryo-EM map. Are Asp368 and ArgH100e side chain density visible in the cryo-EM map? However, I am unable to see any extra density. Also, where is the cavity in Figure 7F?

Response:

We agreed with Reviewer #3 that the densities at some regions particularly the apical regions of the trimer are not well resolved resulted from the disordered conformation including the V1V2 as well as the C4 region. Indeed, the side-chain densities belonging to the Asp³⁶⁸ in CD4 binding loop are not well defined while those of Arg^{H100e} in CDR H3 were fully fitted into its densities, showing its direction pointing towards the CD4 binding loop. Based on our ELISA data, mutation at the Asp³⁶⁸ (D368R) significantly reduced the binding activity of HmAb64, suggesting residue Asp³⁶⁸ plays an important role in HmAb64 recognition. However, this would not affect the results that HmAb64 approached CD4 binding loop of gp120 at an angle from the central axis of the Env trimer to the inward side of the CD4 binding loop.

For the Phe43 cavity, we would like to refer Reviewer to Fig. 6e. This conserved pocket is located at the interface between the inner and outer domains of gp120, named after residue Phe43 of its primary receptor CD4 (in purple sticks).

No structure is shown after fitting the atomic model into the cryo-EM map. Hard to compare map Vs. Model fitting. Additionally, the authors mentioned in the method section that “Due to poor resolution at the regions corresponding to the trimer apex in the final 3D reconstruction map, the V1V2, V3, and C4 (β 20- β 21) domains of the initial gp120 model were truncated before being fitted into the map using UCSF Chimera v1.13.1 71.” Thus, it is essential to show clearly which part is truncated and how was the fitting of the truncated part into the EM map. Definitely, the side chain should be visible at 3.7 Å, and it should be appropriately presented in the figures, at least in the interaction’s region.

Response: We have now labeled clearly where the truncated regions in Supplementary Fig. 6. We also include some side chains in a local density map in Supplementary Fig 5c.

The outer part of the antibodies (dark blue) is not appropriately resolved; It appeared as some dust particles. Thus, it is possible that the outer region of the complex is flexible and difficult to resolve the high-res structure. Generally, that is the case for many of these types of viral protein complexes because they are highly glycosylated (spike-protein, Influenzas immunogen, HIV trimer), and the glycosylated part (the external part of spike, HIV trimer) is challenging to resolve at high resolution. In this current study, we can see that the outer part of the map is less resolved than the inner core part.

Supplementary Figure 3, local resolution calculation shows authors achieved a high-resolution structure at the exterior region (cyan color 2Å), and the core is comparatively high resolution. Generally, it should be the opposite, and the core should be stable and able to reach high res. Generally, the core is more stable and easy to achieve high res, but authors achieved different results. Additionally, the map also shows that Fab density is weaker than the core. These two figures contradict each other and are difficult to understand. So, the authors should clarify this.

Response: We appreciate the Reviewer's careful inspection and kind comments. We agreed that the local resolution mentioned in the region may not be accurately resolved, potentially leading to an overestimation, given its position at the outermost edge of the overall structure. Moreover, this data presented a significant preferred orientation issue, as described in the Methods section, despite the implementation of tilted images. We believe this could be another possibility. To avoid confusion among readers, we have decided to remove this panel from the supplementary figure.

The authors should have mentioned how they calculate the local resolution in the method section. They should try local resolution estimation of unfiltered maps with ResMap.

Response: The local resolution estimation was analyzed using the program in cryoSPARC. We have now removed this panel to avoid confusion.

At 3.7 Å resolution, some bulky side chains should be clear in the density map. Supplemental Figure 3d is the only figure where the authors represented a partial EM map fitted with the model. However, none of the side chains are clearly observed.

Response: We agree with the Reviewer some of the side chains can be assigned and actually our structural descriptions in the Results section were based on the side chains of our model. We have now displayed and labeled some side chains in Fig. 6f and Supplementary Fig. 5c (original Supplemental Figure 3d).

Why did authors collect data at “-” negative defocus values to “+” positive defocus values defocus range of -5.0 to 8.1 μm? Also, the authors collected data using a dose rate of 35.47 e⁻·Å⁻²·s⁻¹ with a total exposure of 1.6 s, for an accumulated dose of 56.76 e⁻·Å⁻² over 40 frames. Thus, per frame exposure time is 0.04. Is this correct?

Response: The effective defocus range should be -0.5 to -2.5 μm, and the per-frame exposure time is 0.04 seconds. We have now correctly described it in Methods.

Reviewers' Comments:

Reviewer #1:

Remarks to the Author:

I thank the authors for engaging with and responding to the comments and feedback from the reviewers.

Line 46: typo – lone 't' should be deleted

Fig. 2c 6535.3 is summarized as tier 2, but is classified as tier 1B as far as I'm aware (<https://www.ncbi.nlm.nih.gov/pmc/articles/PMC2812321/>).

Supplementary Figure 1b. The ZM233.6 neutralization curve is abnormal. I would strongly suggest that the authors repeat this assay if they have not already (I see no mention of replicates in the methods for the larger panel) as the entire signal is coming from essentially two wells. Could the automated liquid handler have potentially failed to add cells or pseudovirus to these two wells?

The authors provide an atypical classification for strains grouping together tier 1B with tier 2 as 'tier 1B/tier 2' (in the text and in Fig 2d). What is the basis for classification as Tier 1B/Tier 2 as opposed to Tier 1B? Furthermore, using percentages of only the neutralized viruses (eg. 'representing 33% of viruses neutralized') is also misleading and is skewed by the nature of the panel. An honest presentation of the neutralization capacity of this monoclonal antibody would be that it is capable of neutralizing three - potentially only two - tier 2 strains.

Table 1. IC50 values should ideally be rounded as the reporting to 3 decimal places suggests a false precision.

Reviewer #2:

Remarks to the Author:

Overall the manuscript is much improved but some areas of my previous critique had not been fully addressed in the actual text of the manuscript. In addition, there are some typos/formatting issues need to be corrected and the manuscript should be checked thoroughly.

Specific issues

Line 317: "This finding supported the long-held belief that DNA immunization can deliver antigens with better conformation compared to the same antigen delivered in the form of a recombinant protein."

This statement needs to be changed to "our long-held belief", while this may be the view of the authors based on one previous paper in rabbits I don't think it is a widespread long-held belief across the HIV vaccine field. On the contrary, there are studies where DNA showed minimal immunogenicity and most HIV immunisation studies recently have used proteins which argues strongly against this being a generally wide held belief. mRNA options obviously hold promise for delivering greater diversity but I do not feel there is a strong enough evidence base presented here for such a strong statement on DNA providing an improved configuration

Line 336: "although the overall percentage of breadth I remains to be moderate"

Previous review comment requested alteration of the word "moderate" to describe breadth to "low". Despite the response to reviewers saying this has been implemented it has not been done, please correct.

Line 342: "At the same time, it also raises the question of how to translate the breadth of neutralizing

antibodies from in vitro assay to real-world protection which warrants further research. The real protection potential of PDPHV can only be demonstrated in an efficacy study.”

Previous review comment was that jumping from these results to a human efficacy study was too great a leap and suggested a more pragmatic next step is discussed instead. The response to reviewers document acknowledges this but the text appears to be unaltered? Also, given the data on controlled NHP challenge studies in the presence of mAbs and the data from human bnAb studies, the first part of the sentence is not really true and seems a bit randomly added here.

Reviewer #3:

Remarks to the Author:

The authors modified several figures and explained most of the reviewer's suggestions. The authors addressed most of the questions, and most of the responses are convincing, although some figure representations need to be corrected.

A couple of minor comments are listed below:

1. The authors showed 2D mesh-like lattice structure and 2D class averages in Supplementary Figure 4B. However, whatever they mentioned as "aggregates", which is not aggregates. It should be ice contamination. The contrast of these particles looks like ice contamination. Also, the image is taken as a snapshot from CrySPARC. It will be better to take the real image without different numbering at the top of the images as well as in class averages. The total number of particles in a particular class and the resolution of that class could be mentioned in figure legends.

2. Yes, the authors added a comparison of gp120 with b12 and gp120 with ab1303 separately. New Supplementary Figures 7 and 8, are added to compare the overall rotation of gp120 and gp41 subunits in the closed form, HmAb64-bound, and b12-bound states (now Fig. 6d. However, in the main text, Supple Fig &g-h. It just added one extra figure in supplementary data files, and no extra values were added to the manuscript. If a new modified figure is added, it should be discussed in the main text (result section), and a discussion should be added on how current antibodies are different from the previous antibodies. Also, if there is a high-resolution structural study, then authors should provide more info, like which amino acid residues are responsible for the interaction. Also, from modified Fig 6D, I am unable to see any open-close forms.

3. The authors answered one of the questions: "We have now labeled clearly where the truncated regions are in Supplementary Fig. 6. We also include some side chains in a local density map in Supplementary Fig 5c." From Suppl figure 5c, side chains could be clearer. It would be better if the authors just represented a couple of chains separately. Like 3-4 helix individually fitted with the atomic model. Not any globular structure, or part of the entire 3D where it is difficult to visualize the side chain fitting. Maybe overall fitting is good but difficult to follow. I suggest just showing 3-4 helix (5-6 amino acids) fitting.

Yes, some side chains are labeled in Fig. 6f, but they are not correctly displayed. The same is true for Supplementary Fig. 5c.

REVIEWER COMMENTS

The authors would like to thank reviewers for their helpful comments and below are our responses.

Reviewer #1:

I thank the authors for engaging with and responding to the comments and feedback from the reviewers.

Line 46: typo – lone ‘t’ should be deleted

Response:

The lone ‘t’ is now deleted in revised manuscript.

Fig. 2c 6535.3 is summarized as tier 2, but is classified as tier 1B as far as I’m aware (<https://www.ncbi.nlm.nih.gov/pmc/articles/PMC2812321/>).

Response:

Neutralizing activity data in the current report were generated by using the VRC 208 virus panel which was developed for a much broader study as published in 2019 <https://doi.org/10.1016/j.str.2018.10.007>. The viruses used in the VRC 208 panel went through a comprehensive assessment and 6535.3 was classified as Tier 1B/2. We have already included this virus as Tier 1B/2 in Fig. 2d and supplementary Fig 1a, and now we further corrected Fig. 1c to mark it as a tier 1B/2 virus.

Supplementary Figure 1b. The ZM233.6 neutralization curve is abnormal. I would strongly suggest that the authors repeat this assay if they have not already (I see no mention of replicates in the methods for the larger panel) as the entire signal is coming from essentially two wells. Could the automated liquid handler have potentially failed to add cells or pseudovirus to these two wells?

Response:

We agree with the reviewer and now removed this virus ZM233.6 from Fig. 2b, 2c, 2d and supplementary Fig 1b.

The authors provide an atypical classification for strains grouping together tier 1B with tier 2 as ‘tier 1B/tier 2’ (in the text and in Fig 2d). What is the basis for classification as Tier 1B/Tier 2 as opposed to Tier 1B? Furthermore, using percentages of only the neutralized viruses (eg. ‘representing 33% of viruses neutralized’) is also misleading and is skewed by the nature of the panel. An honest presentation of the neutralization capacity of this monoclonal antibody would be that it is capable of neutralizing three - potentially only two - tier 2 strains.

Response:

While the original tiered categorization of HIV-1 Env pseudoviruses was introduced in 2010 as the reviewer cited, the numbers of viruses included in the original panel was limited. The VRC 208 panel was subsequently developed to include more pseudoviruses expressing diverse primary HIV-1 Env sequences, and the tier definition was verified/updated based on further characterization. The first use of this VRC 208 panel was in a comprehensive structure study between Nab and Env epitopes

<https://doi.org/10.1016/j.str.2018.10.007>) and John Mascola who was the senior author of 2010 tier definition paper is also a co-author and a key participant for the development of VRC 208 panel.

We have removed percentage description of neutralized viruses. We also changed the number of tier 2 viruses neutralized to two from three as suggested by the reviewer.

Table 1. IC50 values should ideally be rounded as the reporting to 3 decimal places suggests a false precision.

Response:

The IC50 values in Table 1 are now reported with one decimal point.

Reviewer #2 (Remarks to the Author):

Overall the manuscript is much improved but some areas of my previous critique had not been fully addressed in the actual text of the manuscript. In addition, there are some typos/formatting issues need to be corrected and the manuscript should be checked thoroughly.

Specific issues

Line 317: “This finding supported the long-held belief that DNA immunization can deliver antigens with better conformation compared to the same antigen delivered in the form of a recombinant protein.”

This statement needs to be changed to “our long-held belief”, while this may be the view of the authors based on one previous paper in rabbits I don’t think it is a widespread long-held belief across the HIV vaccine field. On the contrary, there are studies where DNA showed minimal immunogenicity and most HIV immunisation studies recently have used proteins which argues strongly against this being a generally wide held belief. mRNA options obviously hold promise for delivering greater diversity but I do not feel there is a strong enough evidence base presented here for such a strong statement on DNA providing an improved configuration

Responses:

We now revised to “our long-held belief” as suggested by the reviewer.

At the same time, we are happy to share the following from Chapter 62 of the classical Stanley Vaccines textbook: “...proteins express from gene-based DNA vaccines are more likely to assume a native conformation, and their expression within cells allows for more native processing and presentation of antigens...” (page 1392, 5th Edition, Vaccines, Saunders, 2008).

Better antigen conformation is different from immunogenicity. DNA vaccine is well known for its low immunogenicity and this is the key reason for our group to promote the heterologous DNA prime and protein boost approach.

Line 336: “although the overall percentage of breadth I remains to be moderate”
Previous review comment requested alteration of the word “moderate” to describe breadth to “low”. Despite the response to reviewers saying this has been implemented it has not been done, please correct.

Responses:

Now we changed ‘moderate’ to “low” in the revised manuscript.

Line 342: “At the same time, it also raises the question of how to translate the breadth of neutralizing antibodies from in vitro assay to real-world protection which warrants further research. The real protection potential of PDPHV can only be demonstrated in an efficacy study.”

Previous review comment was that jumping from these results to a human efficacy study was too great a leap and suggested a more pragmatic next step is discussed instead. The response to reviewers document acknowledges this but the text appears to be unaltered? Also, given the data on controlled NHP challenge studies in the presence of mAbs and the data from human bnAb studies, the first part of

the sentence is not really true and seems a bit randomly added here.

Responses:

We agree that the last revision was not done well. We have now completely revised the paragraph and removed part of it to the end of Discussion.

Reviewer #3 (Remarks to the Author):

The authors modified several figures and explained most of the reviewer's suggestions. The authors addressed most of the questions, and most of the responses are convincing, although some figure representations need to be corrected.

A couple of minor comments are listed below:

1. The authors showed 2D mesh-like lattice structure and 2D class averages in Supplementary Figure 4B. However, whatever they mentioned as "aggregates", which is not aggregates. It should be ice contamination. The contrast of these particles looks like ice contamination. Also, the image is taken as a snapshot from CrySPARC. It will be better to take the real image without different numbering at the top of the images as well as in class averages. The total number of particles in a particular class and the resolution of that class could be mentioned in figure legends.

Response:

We have now modified Supplementary Figure 4B and revised the legend to prevent confusion.

2. Yes, the authors added a comparison of gp120 with b12 and gp120 with ab1303 separately. New Supplementary Figures 7 and 8, are added to compare the overall rotation of gp120 and gp41 subunits in the closed form, HmAb64-bound, and b12-bound states (now Fig. 6d. However, in the main text, Supple Fig &g-h. It just added one extra figure in supplementary data files, and no extra values were added to the manuscript. If a new modified figure is added, it should be discussed in the main text (result section), and a discussion should be added on how current antibodies are different from the previous antibodies. Also, if there is a high-resolution structural study, then authors should provide more info, like which amino acid residues are responsible for the interaction. Also, from modified Fig 6D, I am unable to see any open-close forms.

Response:

We agree with the reviewer and acknowledge the importance of conducting further comparisons between HmAb64, a gp120-based vaccine-induced CD4bs nAb, and currently identified CD4bs nAbs to advance our understanding of subunit-based vaccine design for future studies. However, what we would like to emphasize through those figures, and have mentioned in the main text is how HmAb64 distinguishes itself in recognizing the Env trimer through a different binding mode and approach angle to the CD4 binding site on gp120 compared to other known CD4bs nAbs. This leads to the engagement of epitope regions often not fully exposed in trimer conformations. Specifically the area behind the $\beta 20/\beta 21$ loop, which is typically inaccessible to antibodies in the open occluded or the closed form Env trimer. Here, HmAb64, induced by gp120 immunogens, demonstrates an ability to recognize the Env trimer. Most importantly, it can neutralize about 10% of a panel of viral strains, including tier-2 strains. This suggests that subunit-based immunogens may still hold promise by targeting certain regions potentially buried in the closed form trimer for a subset of viral strains.

For Figure 6d, we have made some modifications to the labeling and revised the legend to make it clear.

3. The authors answered one of the questions: "We have now labeled clearly where the truncated regions are in Supplementary Fig. 6. We also include some side chains in a local density map in

Supplementary Fig 5c." From Suppl figure 5c, side chains could be clearer. It would be better if the authors just represented a couple of chains separately. Like 3-4 helix individually fitted with the atomic model. Not any globular structure, or part of the entire 3D where it is difficult to visualize the side chain fitting. Maybe overall fitting is good but difficult to follow. I suggest just showing 3-4 helix (5-6 amino acids) fitting.

Yes, some side chains are labeled in Fig. 6f, but they are not correctly displayed. The same is true for Supplementary Fig. 5c.

Response:

We have modified Supplementary Fig. 5c to provide a zoomed-in visualization. Additionally, as suggested by the reviewer, we have included additional regions, including HR1 and HR2 of gp41, as well as the α 1 helix of gp120.